# TESSERACT: GRADIENT FLIP SCORE TO SECURE FEDERATED LEARNING AGAINST MODEL POISONING ATTACKS

## ABSTRACT

Federated learning—multi-party, distributed learning in a decentralized environment—is vulnerable to model poisoning attacks, more so than centralized learning. This is because malicious clients can collude and send in carefully tailored model updates to make the global model inaccurate. This motivated the development of Byzantine-resilient federated learning algorithms, such as Krum, Bulyan, FABA, and FoolsGold. However, a recently developed untargeted model poisoning attack showed that all prior defenses can be bypassed. The attack uses the intuition that simply by changing the sign of the gradient updates that the optimizer is computing, for a set of malicious clients, a model can be diverted from the optima to increase the test error rate. In this work, we develop TESSERACT—a defense against this directed deviation attack, a state-of-the-art model poisoning attack. TESSERACT is based on our intuition that in federated learning, certain patterns of gradient flips are indicative of an attack. This intuition is remarkably stable across different learning algorithms, models, and datasets. TESSERACT assigns reputation scores to the participating clients based on their behavior during the training phase and then takes a weighted contribution of the clients. We show that TESSERACT provides robustness against even a white-box version of the attack.

## 1 INTRODUCTION

Federated learning (FL) Smith et al. (2017); Yang et al. (2019) offers a way for multiple clients on heterogeneous platforms to learn collaboratively without sharing their local data. The clients send their local gradients to the parameter server that aggregates the gradients and updates the global model for the local clients to download. FL can be attacked during the training phase by compromising a set of clients that then send maliciously crafted gradients. The attack can be targeted against particular data instances or can be untargeted. The latter brings down the overall accuracy by affecting *all* classes. To counter this threat, a set of approaches has been developed for countering Byzantine clients in FL, *e.g.*, Krum Blanchard et al. (2017), Bulyan Mhamdi et al. (2018), Trimmed Mean and Median Yin et al. (2018), FoolsGold Fung et al. (2020), and FABA Xia et al. (2019). In this work, we use the state-of-the-art (SOTA) untargeted model poisoning attack called a *directed deviation attack*, proposed recently in Fang et al. (2020). *This has been shown to bypass all existing Byzantine-robust aggregation techniques,* e.g.*, Krum, Bulyan, Trimmed mean, and Median.* In our experiments, this attack has been found to decrease the test accuracy from 90% to a low 9% when a DNN is trained using Krum on the MNIST dataset, distributed among 100 clients, 20 of which are under the attacker's control. We describe the relevant details of this attack in Section 2.2.

**Our solution.** We propose a novel defense called TESSERACT against untargeted model poisoning attacks (Figure 1), which uses a stateful model to reduce the contributions by suspicious clients to the global model update. We show that where all prior Byzantine-resilient federated learning approaches fail against the directed-deviation attack of Fang et al. (2020), which is the state-of-the-art untargeted model poisoning attack, TESSERACT is able to recover the test accuracy of the trained model. This benefit applies even when the attack knows the algorithm and all the parameters of our defense, *i.e.*, an adaptive white-box attack. TESSERACT is based on a simple intuition that for a sufficiently small learning rate, as the model approaches an optima in a benign setting, a large number of gradients do not flip their direction with large magnitudes, that is, a degree of inertia is maintained. Our intuition is supported from the analysis we show in Figure 2. We capture this quantitatively in a metric that we propose, called *flip-score*, which is the sum of squares of gradient magnitudes of all parameter updates that suggest a flip in the gradient direction from the previous global update. We find that

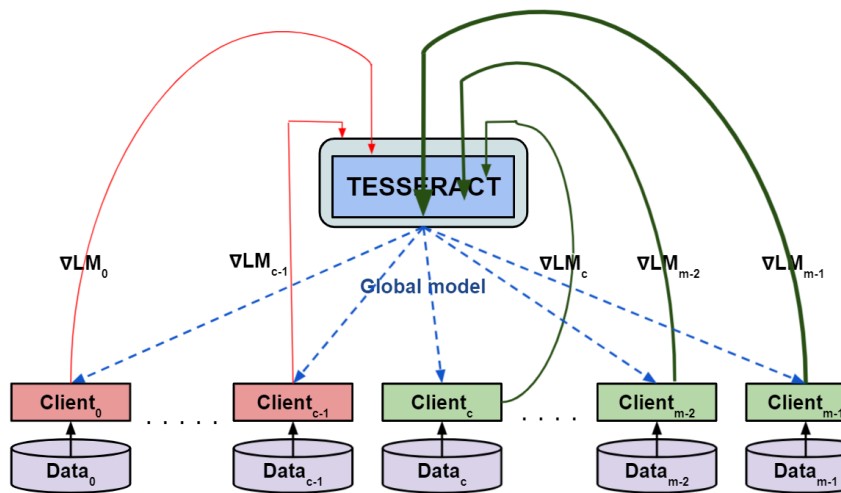

*Figure 1:* TESSERACT*'s architecture where $c$ out of $m$ clients may be malicious and send carefully crafted values of their local models to throw the global model off convergence.* TESSERACT *weights the gradients, received from the clients, by their reputation scores before aggregation, (see varying and thicknesses of arrows from the clients.)*

our intuition and correspondingly the defense TESSERACT also holds for other attacks, *e.g.*, the targeted and untargeted label flipping attacks (Appendix § A.3). However, the other attacks are less damaging, consistent with the observation in Fang et al. (2020). Thus, in the main paper, we focus on the directed deviation model poisoning attack.

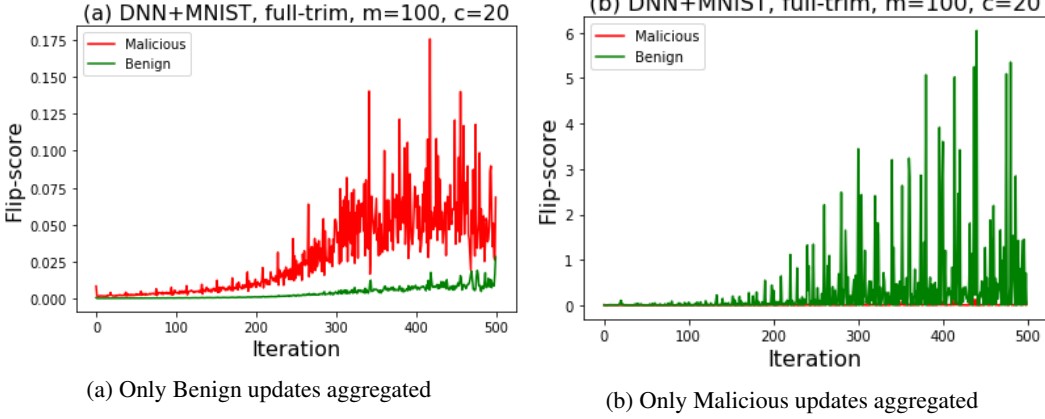

(a) Only Benign updates aggregated

(b) Only Malicious updates aggregated

*Figure 2: The average flip-score of malicious and benign clients over time for a motivating experiment where a DNN is trained on MNIST for 500 iterations (80 benign, 20 malicious clients). In (a), only benign updates were aggregated using FEDSGD, and in (b), only malicious updates were aggregated, depicting the two extreme cases of federated learning in a malicious setting. These results show that when the global model update is benign, the malicious clients send gradients with high flip-scores to deviate the model from convergence. However, when the global model update itself was poisoned, the benign clients send high flip-score gradients for recovery whereas the malicious clients maintain the direction of the already-poisoned model. Thus the intuition is too high flip-scores and too low flip-scores in a coordinated manner are red flags.*

In summary, TESSERACT makes the following contributions.

1. We use our simple intuition to detect malicious clients that attack federated learning using the SOTA attack model Fang et al. (2020). Our intuition is that certain patterns of flips of the signs of gradients across multiple model parameters and across multiple clients are rare under benign conditions.

2. We use a stateful suspicion model to keep the history of every client's activity and use that as a weighting factor in the aggregation. We theoretically prove the convergence of TESSERACT that uses the weighted averaging and establish a convergence rate.

3. We evaluate TESSERACT on DNNs trained on MNIST and FEMNIST, ResNet-18 on CIFAR-10, and GRU on the Shakespeare dataset. We comparatively evaluate our defense against six baselines, including the most recent ones, FABA Xia et al. (2019), FoolsGold Fung et al. (2020), and FLTrust Cao et al. (2020) and show that TESSERACT remains robust even against an adaptive white-box attacker. While several of the existing defenses shine under specific configurations (combination of attacks and datasets/models), TESSERACT is the only one whose protection transfers well across configurations. We release the source code, the attack scripts, the trained models, and the test harness for the evaluation at the anonymized page https://www.dropbox.com/sh/h9ulw6y2f8rzv64/AADe1Sb9PhhCLqclzgZ4xvvJa?dl=0.

The rest of the paper is organized as follows. We describe in Sec 2 the threat model, the SOTA attack, and why all existing Byzantine-resilient federated learning approaches are susceptible. We present TESSERACT's design in Sec 3 and convergence analysis in Sec 4. We describe the baselines and the datasets in Sec 5, and evaluation in Sec 6.

## 2 BACKGROUND

Our simulation of federated learning consists of $m$ clients, each with its own local data, but with the same model architecture and SGD optimizer, out of which $c$ are malicious, as shown in Figure 1. The parameter server assumes that a maximum of $c_{max}$ number of clients can be malicious. The clients run one local iteration, send their gradients (in unencrypted form) to the server, which updates the global model for the clients to download in a synchronous manner.

### 2.1 BYZANTINE-RESILIENT FEDERATED LEARNING

Here we describe the leading defenses briefly, stymied by SOTA untargeted model poisoning attacks.

1. The simplest aggregation technique is **FedSGD** McMahan et al. (2017) that does a simple weighted mean aggregation of the gradients, weighted by the number of data samples each client holds. FedSGD can be attacked by a single malicious client that can send boosted gradients.

2. **Trimmed mean and Median** Yin et al. (2018) aggregate the parameters independently, where one trims $c_{max}$ each at the lower and higher extremes of every parameter and the other takes the median of every parameter update across the gradients received from all the clients. The full-Trim attack Fang et al. (2020) is specifically designed toward these aggregations rules.

3. **Krum** Blanchard et al. (2017) selects one local model as the next global model. The client that has the lowest Euclidean distance from its closest $(m - c_{max} - 2)$ neighbors is chosen as the local model. The full-Krum attack Fang et al. (2020) is tailored to attack Krum.

4. **Bulyan** Mhamdi et al. (2018) combines the above approaches by running Krum iteratively to select a given number of models, and then running Trimmed Mean over the selected ones. The Full-Trim attack is also transferable to Bulyan.

5. **FABA** Xia et al. (2019) iteratively filters out models farthest away from the mean of the remaining models, $c_{max}$ number of times before returning the mean of the remaining gradients.

6. **FoolsGold** Fung et al. (2020) was motivated to defend against poisoning attacks by Sybil clones, and thus, it finds clients with similar cosine similarity to be malicious, penalizes their reputation, and returns a weighted mean of the gradients, weighed by their reputation.

7. **FLTrust** Cao et al. (2020) bootstraps trust in the clients by assuming that the server has access to a clean validation dataset, albeit small, and returns a weighted mean of the gradients weighed by this trust. In our setting, we do not see a realistic method to access such as clean dataset, especially considering the non-iid nature of the local datasets at the clients.

### 2.2 THREAT MODEL: STATE-OF-THE-ART MODEL POISONING ATTACK

Our threat model consists of a scenario where an adversary compromises a fraction of all clients participating in federated learning. We assume that the adversary also has access to the gradient vector sent by the benign clients to the server, and knows what aggregation algorithm the server is running. We focus on the SOTA untargeted model poisoning attack Fang et al. (2020)—a directed deviation attack (DDA) bypassing *all known defenses* [1]. To address generality (more details in Appendix

---

[1]There is no overlap between the authors of this current submission and of the SOTA attack Fang et al. (2020).

§ A.6.2), we have also evaluated TESSERACT on label-flipping attacks in Appendix § A.3. The DDA changes the local models on the compromised worker devices. This change is done strategically (through solving a constrained optimization problem) such that the global model deviates the most toward the *inverse of the direction* along which the benign global model would have changed. Their intuition is that the deviations accumulated over multiple iterations would make the learned global model differ from the benign one significantly.

TESSERACT defends against both variants of the DDA, one specialized to poison Krum (transferable to Bulyan, we call this the *Full-Krum attack*), the other specialized for Trimmed Mean (transferable to Median, we call this the *Full-Trim attack*), respectively. We assume a full-knowledge (white-box) attack where the attackers have access to the current benign gradients. They themselves compute the benign gradients on their local data as well, and thus estimate the benign direction as the average of all benign gradients. This value is stored in a vector $s$ of size equal to the number of model parameters.

$$s(t, \cdot) = sign(\underset{i}{sum}(\nabla LM_i(t, \cdot)),$$

where $\nabla LM_i(t, \cdot)$ is the gradient updates of client $i$.

**Full-Krum attack:** Having estimated $s$, the attackers send gradients in the opposite direction, all with a magnitude $\lambda$, with some added noise to appear different but still maintaining a small Euclidean distance from one another. The upper bound of $\lambda$ is computed in every iteration as a function of $m, c, |P|, GM(t, \cdot), \nabla LM_i(t+1, \cdot)$, where $c$ out $m$ participating clients are malicious, $|P|$ is the number of parameters in the global model $GM$. $\lambda$ is then iteratively decreased until the attackers make sure (using a local estimate) that the parameter server would have chosen the attacked model, using the Krum aggregation technique.

**Full-Trim attack**: The Trim attack while following the same fundamental principle of flipping the gradient direction, attempts to skew the distribution of every parameter $j$ toward the inverse of the direction that $s(t, j)$ suggests in order to attack a mean-like aggregation. It does so by randomly sampling gradient magnitudes from a range that has been computed by the attackers, guaranteed to skew the gradient distribution of every parameter, without appearing as obvious outliers (which would be caught by a method such as Trimmed Mean). Therefore, the attacked gradients here look more diverse than those in the Full-Krum attack. Overall, both forms of the DDA smartly take advantage of the fact that none of the existing aggregation techniques looks at the change in gradient direction to identify malicious gradients in a robust way.

This kind of an attack is an extremely likely scenario in a cross-device as well as a cross-silo federated learning setting, (more details in Appendix § A.6.1) and needs to be defended against.

## 3  DESIGN

TESSERACT uses a reputation-based scheme to compute the aggregation weights of the participating clients. Reputation-based schemes have been widely used in the literature. For example, Kang et al. (2019) and Zhang et al. (2012) make use of reputation score as an incentive mechanism for clients to remain in the system. Fung et al. (2020) and Awan et al. (2021) compute reputation score from the pairwise cosine similarity of the gradients between clients, Cao et al. (2021) does that by computing the cosine similarity of the client gradients with reference to trusted gradients calculated on a clean validation dataset at the server. We compute reputation-score in a different way by using flip-score of the local gradients, as described below. TESSERACT assumes that a maximum of $c_{max}(< \frac{m}{2})$ clients can be malicious. It penalizes $2c_{max}$ clients and rewards the rest in every iteration by an amount $\mathcal{W}(i, t)$ based on their flip-score (described below) and updates their reputation score, where $i$ is the client ID and $t$ is time. We present the pseudocode in Algorithm 1.

$$\mathcal{W}(i, t) = \begin{cases} -(1 - \frac{2c_{max}}{m}), & if \quad penalized. \\ \frac{2c_{max}}{m}, & if \quad rewarded. \end{cases}$$

These values make sure that that the expectation of the reputation score of a client is zero if their flip-scores belong to a uniform random distribution (see Appendix § A.1). Also, this makes recovery difficult as the penalty value is higher than the reward as $c_{max} < \frac{m}{2}$ and allows us to be conservative. The reputation score of a client $i$ is initialized and updated as follows -

$$RS(i, 0) = 0.$$
$$RS(i, t) = \mu_d RS(i, t-1) + \mathcal{W}(i, t), \quad t > 0.$$

where $0 \leq \mu_d \leq 1.0$ is the decay parameter, and $RS$ is the reputation score. A low $\mu_d$ gives more importance to the present flip-score and a high $\mu_d$ gives significant importance to the past performance of a client. This decay operation also helps cap the maximum and minimum reputation score (with $\mu_d < 1$, for more details, see Appendix § A.1). We normalize the reputation score using softmax to do a weighted mean aggregation, but a user can use this to directly filter out the suspicious gradients and use an aggregation rule of one's choice. We halve the reputation score of every client if any of the values grows so large that the softmax operation causes an overflow. This is motivated by the fact that the reputation score is a cumulative quantity, and can potentially keep increasing with time in an unbounded manner.

**Flip-score.** We compare the present gradients $\triangledown LM_i(t+1, \cdot)$ sent by local model $i$ with the gradient direction of the global model at time $t$, $s_g(t, \cdot) = sign(GM(t, \cdot) - GM(t-1, \cdot))$. We define flip-score as the sum of square of the gradient magnitudes of all parameters that experience a change in their gradient direction, that is,

$$FS_i(t+1) = \sum_{j=0}^{|P|-1} (\triangledown LM_i(t+1,j))^2 (sign(\triangledown LM_i(t+1,j)) \neq s_g(t,j)), \quad (1)$$

where $|P|$ is the total number of parameters in the model being trained. A low flip-score thus suggests that the gradient updates are approximately in the same direction as the previous iteration. In contrast, a high flip-score suggests a deviation from the previous update. This could mean either a large number of parameters have flipped direction, or a small number of parameters have flipped direction with large magnitudes, or both. As observed in Figure 2, if the previous global update was benign, a malicious client will tend to have a high flip-score. However, if the previous update itself was poisoned, the flip-score of benign clients will be high and those of malicious clients will be low. Therefore, we penalize $c_{max}$ number of clients at either end of the current flip-score distribution, as also done by Yin et al. (2018), but we do this in the context of gradient values per parameter. This allows our system to have a higher detection coverage irrespective of whether the global model was poisoned in one iteration or not. Since we allow redemption, and make sure that in a random penalization scheme, the expectation of reputation score of a benign node approaches zero, we do not unnecessarily penalize benign clients with low flip-scores. Also, since we do not use any hard threshold for detecting attacks, and identify only the extreme ends of flip-score distribution in every iteration as malicious, we do not prevent low flip-score moves. At the time of convergence, when most of the clients will favor low flip-score moves, such moves will be allowed even after ignoring the gradients at the extreme ends.

---

**Algorithm 1** Federated learning with TESSERACT

---

**Output**: Global model $GM(t+1, \cdot)$
**Input**: Local model updates $w = \triangledown LM_i(t+1, \cdot)$
**Parameters**: $m$, $c_{max}$, $\mu_d$

**0 :** Initialize reputation $RS_i(0) = 0$ for every client $i$
**1 :** Initialize global direction $s_g(0)$ to a zero vector
**2 :** for every client $i$ **compute flip-score**:
**3 :** $\quad FS_i(t+1) = \sum_{j=0}^{|P|-1} (\triangledown LM_i(t+1,j))^2 (sign(\triangledown LM_i(t+1,j)) \neq s_g(t,j))$
**4 :** Penalize $c_{max}$ clients on either end of FS spectrum as: $RS(i, t+1) = \mu_d RS(i,t) - (1 - \frac{2c_{max}}{m})$
**6 :** **Reward** the rest of the clients as: $RS(i, t+1) = \mu_d RS(i,t) + \frac{2c_{max}}{m}$
**7 :** **Normalize** reputation weights: $W_R = \frac{e^{RS}}{\sum e^{RS}}$
**8 :** **Aggregate** gradients: $\triangledown GM(t+1, \cdot) = w^T W_R$
**9 :** Update global direction: $s_g(t+1, \cdot) = sign(\triangledown GM(t+1, \cdot))$
**10:** **Update** global model and broadcast: $GM(t+1, \cdot) = GM(t, \cdot) + \triangledown GM(t+1, \cdot)$

---

## 4  CONVERGENCE ANALYSIS

We make the following assumptions on $LM_k$, part of which are adapted from Li et al. (2020).

1. Assumption #1: $LM_k$ are all L-smooth, that is, for all $v$ and $w$, $LM_k(v) \leq LM_k(w) + (v - w)^T \triangledown LM_k(w) + \frac{L}{2} \|v - w\|_2^2$.

2. Assumption #2: $LM_k$ are all $\mu$-strongly convex, that is, for all $v$ and $w$, $LM_k(v) \geq LM_k(w) + (v - w)^T \nabla LM_k(w) + \frac{\mu}{2}\|v - w\|_2^2$.
3. Assumption #3: Let $\xi_t^k$ be sampled uniformly at random from the local data of the $k - th$ client, then the variance of stochastic gradients of each client is bounded, that is, $\mathbb{E}\|\nabla LM_k(\mathbf{w}_t^k, \xi_t^k) - \nabla LM_k(\mathbf{w}_t^k)\|^2 \leq \sigma_k^2$ for $k = 1, 2, ...m$.
4. Assumption #4: The expected squared norm of stochastic gradients is uniformly bounded, that is, $\mathbb{E}\|\nabla LM_k(\mathbf{w}_t^k, \xi_t^k)\|^2 \leq G^2$ for $k = 1, ...m$, and $t = 0, ..T - 1$.
5. Assumption #5: Within $K$ iterations, the reputation score of malicious clients drop at least by $\delta_{mal}$, and reputation score of benign clients increase at least by $\delta_{ben}$, that is, $|RS_{mal}^t - RS_{mal}^{t-K}| \geq \delta_{mal}$ and $|RS_{ben}^t - RS_{ben}^{t-K}| \geq \delta_{ben}$, for $t = 0, ..T - 1$.

Assumptions #1 and #2 are standard and apply to $l_2$-norm regularized linear regression, logistic regression, and softmax classifier. Assumptions #3 and #4 have been also made by Zhang et al. (2012); Stich (2018); Stich et al. (2018); Yu et al. (2019). In our problem setting, Assumption #3 claims that the gradient with a subset of local data is bounded from the gradient with whole batch for both malicious and benign clients. Assumption #5 claims that after every $K$ iterations, our algorithm will have better ability to distinguish the malicious clients.

Following the Theorem 1 in Li et al. (2020), after simplifying for our case in which the clients communicate with the parameter server in every iteration, we derive this convergence (details in Appendix § A.4). Let $GM^*$ and $LM_k^*$ be the minimum value of $GM$ and $LM_k$ respectively, then:

$$\mathbb{E}[GM(\mathbf{w})_T] - GM^* \leq \frac{2}{\mu^2} \cdot \frac{L}{\gamma + T}(\sum_{k=1}^{m} p_k^2 \sigma_k^2 + 6L\Gamma + 8G^2 + 8G^2 \sum_{k=1}^{c} p_{k,0} + \frac{\mu^2}{4}\|w_0 - w^*\|^2).$$

where $L, \mu, G$ are defined above, $\gamma = max\{8\frac{L}{\mu}, 1\}$, $\eta_t = \frac{2}{\mu(\gamma+t)}$, $T$ is the total number of iterations, $c$ is the number of malicious clients, $p_{k,0}$ is the initial weight for malicious clients, and $\Gamma = GM^* - \sum_{k=1}^{m} p_k LM_k^*$, that effectively quantifies the degree of non-iidness according to Li et al. (2020). This shows that a weighted mean aggregation is guaranteed to converge in federated learning. The converge speed is $O(\frac{1}{T})$. The increase of number of malicious clients $c$ increases the bound.

## 5 IMPLEMENTATION

We have simulated the federated learning on a single machine with a Tesla P100 PCIe GPU with 16GB memory, using PyTorch, with as many clients as can be handled by our machine. The data was distributed with a non-IID bias of 0.5 (default), except for Shakespeare where the data was distributed sequentially among the clients and FEMNIST where the data was distributed by the writer. In our simulation, all clients run one local iteration on a batch of its local data before communicating with the parameter server in a synchronous manner. The clients sample their local data in a round-robin manner, send their local gradients to the parameter server, and download the updated global model before running the next local iteration. The malicious clients attack every iteration of training, we assume: $c = c_{max}$.

**Baselines and datasets**. The baseline aggregation rules used are Krum, Bulyan, Trimmed Mean, and Median. We also compare TESSERACT with the recent defense techniques of FABA Xia et al. (2019), FoolsGold Fung et al. (2020), and FLTrust Cao et al. (2020). We evaluate TESSERACT on 4 different datasets (Table 1). The DNN trained on MNIST has 2 conv layers with 30 and 50 channels respectively, each followed by a $2 \times 2$ maxpool layer, then by a fully connected layer of size 200, and an output layer of size 10. We use a constant learning rate, except for CIFAR-10 where we start with zero, reach the peak at one-fourth of the total iterations, and slowly get down to zero again. The CNN trained on the FEMNIST dataset follows the same network architecture as Caldas et al. (2019).

## 6 EVALUATION

### 6.1 MACRO EXPERIMENTS

In Table 2, we compare the test accuracy achieved by various aggregation techniques in benign and malicious conditions. We do not claim to provide optimal model architectures that can achieve the best test accuracy, but we provide fair comparison of the all the defenses on the same model with the same training parameters. For Shakespeare, which is an NLP dataset, we report the test loss; for all others, we report test accuracy. The final reported test loss value does not capture the training dynamics, which can be observed in Figure 3 for the more damaging Full-Krum attack. We have not shown the test loss curve for Krum aggregation because of the large loss values here,

Table 1: *Datasets with the number of classes ($n_c$) and training samples ($n_s$), the models and model parameters ($P$), training rounds ($n_r$), batch_size ($b$), learning rate ($lr$), total number of clients ($m$), number of malicious ones ($c$), and decay parameter ($\mu$) used in* TESSERACT. * *variable learning rate, peaks at 0.1.*

| Dataset | $n_c$ | $n_s$ | Model | P | $n_r$ | b | lr | m | c | $\mu$ |
|---|---|---|---|---|---|---|---|---|---|---|
| MNIST | 10 | 60k | DNN | 0.27M | 500 | 32 | 0.01 | 100 | 20 | 0.99 |
| CIFAR-10 | 10 | 50k | ResNet-18 | 5.2M | 2000 | 128 | 0.1* | 10 | 2 | 0.99 |
| Shakespeare | 100 | - | GRU | 0.14M | 2000 | 100 | 0.01 | 10 | 2 | 0.99 |
| FEMNIST | 62 | 805k | DNN | 6.6M | 2000 | 32 | 0.1 | 35 | 7 | 0.99 |

Table 2: *Attack impact - Test accuracy for Directed Deviation model poisoning attacks (Full-Krum; Full-Trim), on different datasets with $c/m = 0.2$. For the Shakespeare dataset, test loss has been reported. We verify the damaging impact of the Full-Trim attack on mean-like aggregations (FedSGD, Trimmed mean, Median) and Full-Krum attack on Krum-like aggregations (Krum, Bulyan). We also observe that the existing defenses—FABA, FoolsGold, and FLTrust—seem to defend against this attack in some cases, and fail in others, whereas* TESSERACT *consistently shines in all cases.*

| Attack | Defense | Test accuracy (%) / Test loss (only for Shakespeare) | | | |
|---|---|---|---|---|---|
| | | MNIST+ DNN | CIFAR-10+ ResNet-18 | Shakespeare+ GRU | FEMNIST+ DNN |
| None | FedSGD | 92.45 | **71.17** | **1.62** | 83.60 |
| | TESSERACT | **92.52** | 66.92 | 1.64 | 83.58 |
| | FABA | 91.77 | 69.94 | 1.76 | 82.69 |
| | FoolsGold | 91.20 | 70.71 | 1.63 | **83.80** |
| | FLTrust | 87.70 | 68.08 | **1.62** | 82.72 |
| Full-Krum | FedSGD | 82.97 | 39.68 | 1.62 | 29.87 |
| | Krum | 8.92 | 9.81 | 11.98 | 5.62 |
| | Bulyan | 10.14 | 13.24 | 9.23 | 9.91 |
| | TESSERACT | **87.73** | 61.26 | 1.64 | **80.19** |
| | FABA | 86.99 | 55.96 | 1.75 | 55.61 |
| | FoolsGold | 47.12 | 42.28 | **1.63** | 0.07 |
| | FLTrust | 82.50 | **65.25** | 1.67 | 79.53 |
| Full-Trim | FedSGD | 65.25 | 47.32 | 1.74 | 32.34 |
| | Trim | 36.36 | 55.25 | 3.28 | 13.03 |
| | Median | 28.37 | 50.54 | 3.30 | 45.6 |
| | TESSERACT | 90.55 | 67.65 | 1.66 | 82.51 |
| | FABA | **91.84** | 67.31 | **1.64** | 79.66 |
| | FoolsGold | 91.61 | **69.24** | 1.66 | **83.09** |
| | FLTrust | 34.20 | 64.23 | 1.68 | 79.28 |

but have pushed the results to Appendix § A.3 in Figure 9. We see that TESSERACT is the winner or 2nd place finisher in 7 of the 12 cells (benign + two attacks × 4 datasets). This on the surface appears to be not very promising, till one looks deeper. The baseline protocols that finish first in one configuration fare disastrously in other configurations, indicating that they are tailored to specific attacks or datasets (whether by conscious design or as an artifact of their design). For example, FoolsGold does creditably for the Full-Trim attack but is vulnerable against the Full-Krum attack. Averaging across the configurations, it appears FABA is the closest competitor to TESSERACT. We observe that FABA, although it performed well for $c/m = 0.2$, failed to defend when the number of attackers grew to $c/m = 0.3$, evident from the results in Figure 3(b). We have used F-MNIST, Ch-MNIST, and Breast cancer Wisconsin Dataset here to show the effect on diverse datasets. As the number of attackers increases, the mean starts to shift more toward them. One false positive detection by FABA can cause it to trim out a benign local update, which causes the mean to shift further toward the malicious updates iteratively, and fail. On the other hand, TESSERACT is guaranteed to defend against the attack as long as $c \leq c_{max}$ and $c < \frac{m}{2}$. We find empirically (result not shown) that FABA degrades fast, and much faster than TESSERACT, when the fraction of malicious clients increases. Whereas FABA fails at $c/m = 0.3$, we show in Figure 7(a) that TESSERACT remains stable until $c/m = 0.45$. We also show in Figures 7(b) and (c) that TESSERACT remains robust across a wide range of non-IID bias, that is 0.1 to 0.8. TESSERACT and FABA require an estimate of the upper

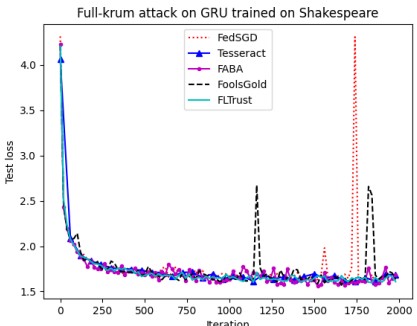 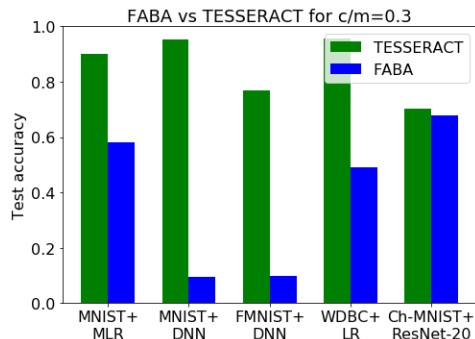

*Figure 3: Left shows the test loss curve comparing* TESSERACT *with the benchmark aggregation algorithms against the Full-Krum attack. We see that the attack generates sporadic spikes in training, best handled by* TESSERACT, *evident from its smooth test loss curve. Right shows the comparison of FABA and* TESSERACT *across diverse datasets for* $c/m = 0.3$. *FABA begins to fail with a higher fraction of malicious clients, while* TESSERACT *remains robust.*

*Table 3: Fraction of malicious or benign clients allotted weights above 1e-4 averaged over 500 iterations. The weakness of FoolsGold and FLTrust stems in part from the fact that they assign low weight to a significant fraction of benign clients for effective detection.*

| Defense | Mal/ Ben | Benign | Full-trim | Full-krum |
|---|---|---|---|---|
| FoolsGold | $n_{ben}$ | 0.29 | 0.30 | 0.10 |
| | $n_{mal}$ | - | 0.00 | 0.64 |
| FLTrust | $n_{ben}$ | 0.48 | 0.45 | 0.49 |
| | $n_{mal}$ | - | 0.52 | 0.63 |
| TESSERACT | $n_{ben}$ | 0.75 | 0.75 | 0.63 |
| | $n_{mal}$ | - | 0.00 | 0.08 |

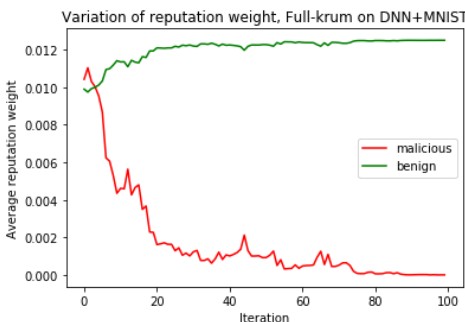

*Figure 4: Clients' reputation weights against time with* TESSERACT, *with higher weights to benign clients, as malicious ones tend to 0.*

bound of number of malicious clients, $c_{max}$ to be known. FoolsGold and FLTrust, on the other hand, make use of cosine similarity among clients, and with a trusted cross-check dataset at the server, respectively, to identify suspicious clients. We have found that both of these techniques unnecessarily penalize many benign clients and assign them a zero weight in order to conservatively defend against an attack, as can be seen in Table 3. This can have a significantly negative impact on a practical system where one wishes to learn from data that the different clients hold locally. On the other hand, TESSERACT allows all clients to contribute to the global model update that do not have a large negative reputation score. Figure 4 shows the evolution of the average reputation score of malicious and benign clients. When benign and malicious clients start from the same region, the weights of malicious clients decrease with time tending to zero, while those of the benign clients increase with time.

## 6.2 ADAPTIVE ATTACK

Having shown the performance of TESSERACT against the above attacks, we proceed to analyze an adaptive attack scenario, *i.e.*, one where the attacker has full knowledge of TESSERACT, including the dynamic value of the cutoff flip-scores. Thus, at iteration $t + 1$, the attacker knows $FS_{low}(t)$ and $FS_{high}(t)$ beyond which the clients were penalized at iteration $t$. The Adaptive-Krum attack first computes the target malicious gradients at $t+1$, and if its flip-score goes above $FS_{high}(t)$, it reverses the attack on its less important parameters, *i.e.*, parameters that would have had low magnitude updates without attack. It does this by replacing 5% of the attacked parameters, at a time, with their benign values until the flip-score is brought down to ensure stealth. Since the stealthy attack will be less powerful than the original intended attack, the global model will only be partially poisoned, and the attackers are not expected to occupy the lower spectrum of the flip-score. This has been verified

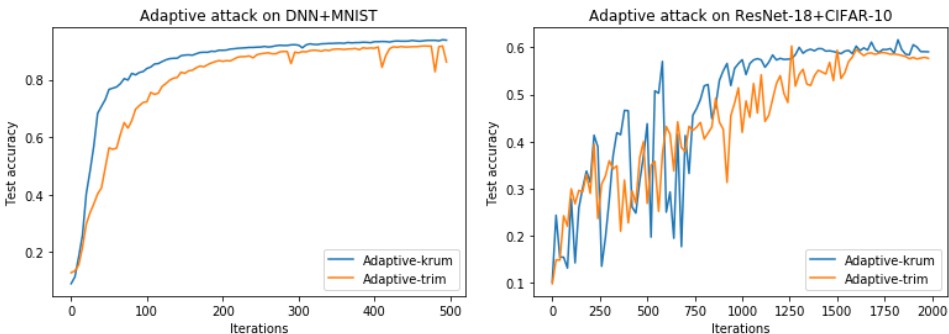

*Figure 5: Performance of* TESSERACT *against adaptive white-box attacks, specifically designed to attack* TESSERACT, *evaluated on MNIST and CIFAR-10. We observe a significant improvement in test accuracy, compared to the base case impact of Full-Krum on Krum and Full-Trim on FedSGD, as reported in Table 2.*

in our experiments as well. All the malicious clients send these attacked parameters with some added randomness in order to support one other. We observe a trade-off between stealth and attack impact in this case, as can be seen in Figure 5.

The Adaptive-Trim attack is a smarter and collaborative attack. Its target is to generate attacked gradients $v_i$, $i = 0, 1, 2, \cdots c - 1$. It first computes the Full-Trim target attack $u_i$ for every malicious client $i$. Then, it initializes $v_0$ to $u_0$ and modifies it, until $v_0$ generates a flip-score that is less than $FS_{high}(t)$. Client $i = 1$ then updates its target attack from $u_1$ to $v_1 = u_1 + (u_0 - v_0)$ in order to compensate for a sub-optimal attack created by $i = 0$ because of the flip-score-evasion constraint. Thus, a client may not necessarily find a solution if the target grows and the constraints are hard to solve for. The attacker hopes that $\sum_{i=0}^{c-1} u_i = \sum_{i=0}^{c-1} v_i$. The performance of TESSERACT against this adaptive white-box attack is shown in Figure 5. We find that the adaptive attacks are not very effective against TESSERACT. The benign accuracy for the two datasets are 92.45% and 71.17%. This happens due to multiple reasons: 1) the attack loses in strength while trying to gain in stealth, 2) the attackers need not be allotted equal reputation weights, so the weighted sum of the attacked gradients $v$ do not match with the weighted sum of the target attack $u$, 3) the flip-score distribution is dynamic, as can be seen in Figure 2, and changes from time $t$ to $t + 1$, and when it decreases in consecutive iterations by a significant amount, the attackers can still be blocked.

We have also created a more knowledgeable attacker, by modifying the above constraint with a weighted sum, weighted by the reputation scores. We have evaluated this attack in Figure 8 in Appendix § A.3. TESSERACT is effective against this attack too, for reasons (1) and (3) given above.

## 7 DISCUSSION AND CONCLUSION

We have presented TESSERACT, a secure parameter server for federated learning robust to model poisoning attacks. TESSERACT uses a stateful algorithm to allocate reputation scores to the participating clients to lower the contribution of maliciously behaving clients. We define malicious behavior using a metric, flip-score, which when very high or low, captures attacks that try to divert the global model away from convergence. This makes TESSERACT a robust defense, no matter when it is instantiated, although we recommend enabling TESSERACT right from the start. TESSERACT can also be used to just filter out clients based on their reputation, so that an aggregation of the user's choice can be used. We evaluate the benefits of TESSERACT compared to the fundamental FL aggregation FedSGD and state-of-the-art defenses, namely Krum, Bulyan, FABA, FoolsGold, and FLTrust. We evaluate using full knowledge untargeted model poisoning attacks that have recently been found to be most damaging against FL. We find that different existing defenses shine under specific combinations of attacks and datasets/models. However, TESSERACT provides transferable defense with accuracy competitive with individual winners under all configurations. Further, TESSERACT holds up better than its closest competitor, FABA, when the fraction of malicious clients increases (beyond 20%). Finally, an adaptive white-box attacker with access to all internals of TESSERACT, including dynamically determined threshold parameters, cannot bypass its defense. All of our evaluation is limited to a synchronous setting with no gradient encryption, as gradient encryption is less relevant in a

cross-device scenario due to its high computational overhead Zhang et al. (2020). However, these form logical avenues for our future work.

## 8 REPRODUCIBILITY STATEMENT

All the details for the datasets and models used along with the implementation of the adaptive attack is given in Section 5. To help with the reproducibility of the experiments, we make available all of our code, the trained models, the training hyperparameter values, and the raw results from our testing. These are all to be found at the anonymized link https://www.dropbox.com/sh/h9ulw6y2f8rzv64/AADe1Sb9PhhCLqclzgZ4xvvJa?dl=0. The code has also been uploaded as a supplementary file in the current submission.

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

# A    APPENDIX

## A.1    PENALTY AND REWARD VALUE

**Penalty and reward selection.** Our design policy penalizes $2c_{max}$ out of $m$ clients in every iteration. Considering a completely benign scenario, we want the expected value of the reputation score of a client that has been penalized $e$ fraction of times to be zero, where $e = \frac{2c_{max}}{m}$. Let a client $i$ be penalized $en$ number of times in $n$ iterations. There are $\binom{n}{en}$ ways to select the iterations where the client is penalized. After $n$ iterations, the reputation score of client $i$ is given by:

$$RS(i,n) = \sum_{t=0}^{n} \mu_d^{n-t} \mathcal{W}(i,t). \tag{2}$$

where $\mathcal{W}(i,t)$ is a sequence of penalty and reward over time. The expected value of this reputation score over all possible sequences $j \in \binom{n}{en}$ is

$$\mathbb{E}_j[RS(i,n)] = \frac{1}{\binom{n}{en}} \sum_j RS(i,n)$$

$$= \frac{1}{\binom{n}{en}} \sum_j \sum_t \mu_d^{n-t} \mathcal{W}(i,t)$$

$$= \frac{1}{\binom{n}{en}} \sum_t \mu_d^{n-t} \sum_j \mathcal{W}(i,t)$$

$$= \frac{1}{\binom{n}{en}} \sum_t \mu_d^{n-t} (-(pen\binom{n}{en}) + (r(1-e)n\binom{n}{en}))$$

Our setting with $r = \frac{2c_{max}}{m}$ and $p = 1 - r$ makes the above quantity to be zero thus ensuring that its expected reputation score increment is zero. This proof assumes that it is a random process through which (benign) clients generate their flip scores. Thus, if a subset of clients are penalized less than $\frac{2c_{max}}{m}$ of times, they are expected to have a net neutral reputation score.

**Upper and lower bound of reputation score.** From the above expression, it is obvious that if $\mu_d = 0$, $-p \leq RS \leq r$. When $0 < \mu_d < 1$, the upper and lower bounds can be computed by assuming that a client was rewarded or penalized respectively in every iteration. Assuming that the number of iterations tends towards infinity, equation (1) forms an infinite geometric sequence, that can be solved to obtain $\frac{-p}{1-\mu_d} \leq RS \leq \frac{r}{1-\mu_d}$. It should be noted that these reputation scores are normalized using softmax to compute the reputation weights. If the absolute value of the lower bound is not large enough (if $\mu_d$ is set to be too small), then even after perfect detection, a malicious client can still have a significant reputation weight after softmax normalization. If $\mu_d$ is set to a value closer to 1, then the absolute value of the lower and upper bounds increase, bringing down the contribution of malicious clients to almost zero. At the same time, redemption becomes difficult for a client in this case. This trade-off needs to be kept in mind when setting the decay parameter. We have used $\mu_d = 0.99$ in our experiments in order to remain conservative and make recovery difficult for a client that has been penalized a lot of times. However, this is a design parameter that the user can decide.

## A.2    LABEL FLIPPING

The attack that we target, "the directed-deviation attack" has been shown to be the most powerful attack in federated learning Fang et al. (2020), and specifically claims to be more effective than state-of-the-art untargeted data poisoning attacks for multi-class classifiers, that is, label flipping attack, Gaussian attack, and back-gradient optimization based attacks Muñoz-González et al. (2017). They show that the existing data poisoning attacks are insufficient and cannot produce a high testing error rate, not higher than 0.2 in the presence of byzantine-robust aggregation techniques (Krum, trimmed mean, and median).

We observe that both state-of-the-art targeted and untargted label flipping attacks are not powerful enough on the CIFAR-10 and FEMNIST datasets and have neglible damage. The attacks do have some

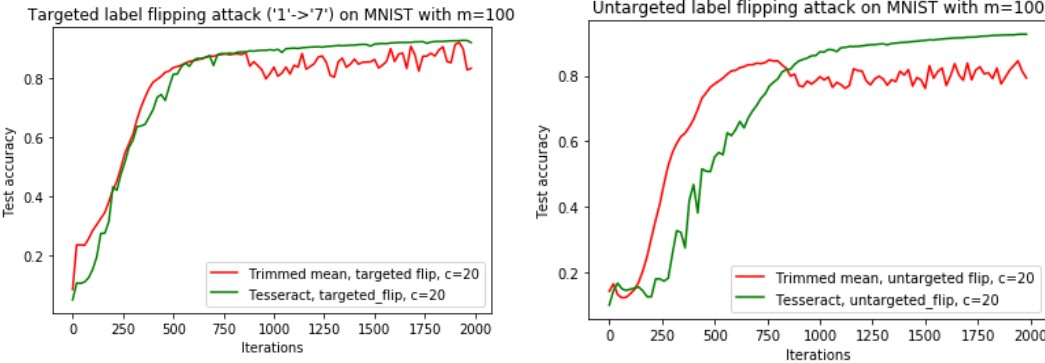

*Figure 6:* TESSERACT*'s Performance against the targeted and untargeted label flipping attacks on the MNIST dataset. We observe that the attacks have some damage on the model, but Tesseract is able to remedy this for both attacks.*

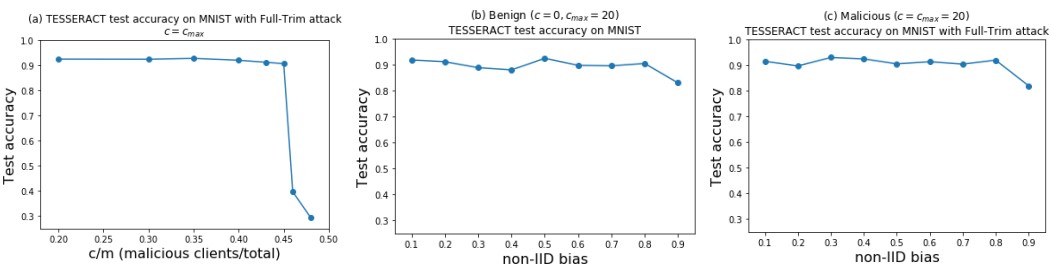

*Figure 7: Figure (a) shows the performance of* TESSERACT *on MNIST with increasing* $c$*. We see that* TESSERACT *is stable across a large range and breaks only above* $c = 0.45$ *which is close to the theoretical limit of* $c = 0.5^-$*. Figures (b) and (c) show the test accuracy of* TESSERACT *on MNIST dataset distributed with varying non-IID bias across 100 clients in benign and malicious cases respectively.* TESSERACT *can be seen to be robust enough for a wide range of bias, 0.1–0.8, with a small dip in test accuracy occurring at* $bias = 0.9$*.*

damaging impact on the MNIST dataset, but when TESSERACT is used, the damage is completely mitigated. Thus, we verify the claims from Fang et al. (2020) and show that TESSERACT's intuition is general enough to counteract both the more powerful "directed-deviation attacks" and the weaker state-of-the-art data poisoning attacks.

### A.3 ADDITIONAL EXPERIMENTS

Here, we provide additional evaluation of TESSERACT in two specific situations. We stress-test it first by subjecting it to a higher number of malicious clients to find the breaking point of TESSERACT, when trained on MNIST dataset in the presence of Full-trim attack. We assume that the number of compromised clients is still not greater than $c_{max}$, and to that end, we set $c = c_{max}$. Since TESSERACT requires $c_{max} < \frac{m}{2}$, we have swept $c$ upto 49 where $m$ was fixed at 100. We observe in Figure 7(a) that TESSERACT is stable upto $c/m = 0.45$ whereas the rest of the defense techniques broke below $c/m = 0.30$ as can be seen in Table 2 and Figure 3 with $c = c_{max}$ set for all the defense techniques that require a knowledge of $c_{max}$.

Figures 7(b) and (c) show the performance of TESSERACT on MNIST dataset distributed among 100 clients with varying degrees of non-IIDness. We observe that, except for the extreme case of $bias = 0.9$, TESSERACT remains exceptionally stable.

Here, we describe the mathematical formulation of the adaptive-attacks. Full-Krum attack finds a vector of gradients $u$ by solving an optimization problem described in Fang et al. (2020), and every malicious client $i$ would send $u$ with an additional noise to appear different. Full-Trim attack solves a different optimization problem to also come up with a vector of gradients $u$ to which every malicious

client $i$ would add some noise to obtain $u_i$. The problem statement in our (two) derived versions of the above (two) attacks, namely Adaptive-Trim and Adaptive-Krum, is To find a set of vectors

$$v_i, i = 0, 1, 2, ..., c-1$$

where $c$ is the number of malicious clients. Here, $v_i$ is the vector of gradients with size equal to the number of model parameters, each satisfying the constraint -

$$FS(v_i) < FS_{high}(t)$$

that is, every computed vector should have a flip-score lower than the cut-off flip-score according to the adversary's knowledge, such that

$$\sum_{i}^{c-1} v_i = \sum_{i=0}^{c-1} u_i$$

where $u_i$ were determined by the adversary originally as a valid solution to the Full-Krum and Full-Trim optimization problems. We solve this problem as described in Section 6.2. In short, we initialize $v_i$ to some target value, and then undo the attack on "less important parameters" until the flip-score constraint is just met, and send the computed $v_i$ for aggregation. $v_0$ is initialized to $u_0$, and then updated until the flip-score constraint is satisfied. The difference $u_0 - v_0$ is added to $u_1$ which now becomes the initial value of $v_1$ and so on. The results have been described and analyzed in Section 6.2.

We formulate an even stronger attack where the adversary also has the knowledge of its own reputation score ($W_R$) in order to come up with attacked gradients with better chances of success. We call this a "Weighted-Adaptive-Trim" attack. The modified constraint is

$$\sum_{i}^{c-1} W_{R,i} v_i = \sum_{i=0}^{c-1} W_{R,i} u_i$$

TESSERACT successfully defends against this attack when evaluated on MNIST, as can be seen from the experimental result in Figure 8. With TESSERACT, the test accuracy reaches 90% while in the baseline case, it only reaches 58%. For context, without any attack, the model reaches accuracy of 92.45%.

Figure 9 shows the devastating impact of Full-Krum attack on Krum aggregation with the Shakespeare NLP dataset. This was not shown in Figure 3(a) because the abnormally high loss values makes the comparison between benchmarks with smaller loss values difficult. All defenses partially mitigate the impact of this attack, but the greatest benefit is obtained from TESSERACT.

## A.4 CONVERGENCE ANALYSIS

Let the $k$-th client hold $n_k$ training data batches: $x_{k,1}, ... x_{k,n_k}$. The local objective function $LM_k(\cdot)$ is given by

$$LM_k(\mathbf{w}) = \frac{1}{n_k} \sum_{j=1}^{n_k} l(\mathbf{w}; x_{k,j}),$$

where $l(\cdot; \cdot)$ is the specified loss function for each client.

The global objective function is defined as

$$GM_{k,t}(\mathbf{w}) = \sum_{k=1}^{m} p_{k,t} LM_k(\mathbf{w}).$$

The global model is updated as

$$\mathbf{w}_{t+1}^k = \mathbf{w}_t^k - \eta_t \sum_{k=1}^{m} p_{k,t} \nabla LM_k(\mathbf{w}_t^k),$$

where $p_{k,t} = softmax(RS_{k,t})$ is the softmax of reputation score of client $k$ at time $t$.

We update the weights by averaging the weights from selected clients $\bar{\mathbf{w}}_t = \sum_{k=1}^{m} p_{k,t} \mathbf{w}_t^k$. For convenience, we also define $g_t = \sum_{k=1}^{m} p_{k,t} \nabla LM_k(\mathbf{w}_t^k, \xi_t^k)$, where $\xi_t^k$ is the selected local data.

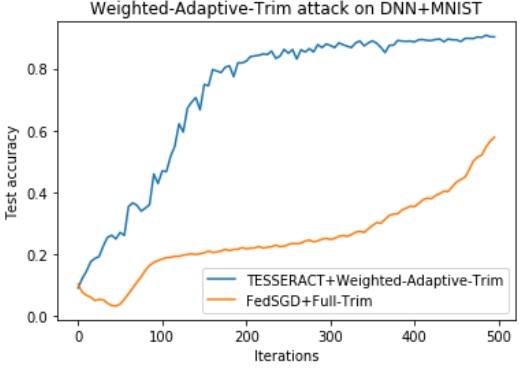

*Figure 8: The figure shows the test accuracy of* TESSERACT *when evaluated on MNIST dataset under the default conditions with $m = 100, c = 20$ where the adversary launches the Weighted-Adaptive-Trim attack on the system, compared with the baseline performance of FedSGD against the Full-Trim attack.* TESSERACT *successfully defends against this attack to achieve a 90% accuracy.*

*Figure 9: The figure shows the impact of the Full-Krum attack on Krum aggregation on the Shakespeare NLP dataset as compared to the other aggregation techniques. It has a devastating impact which is mitigated by all defenses and most effectively by* TESSERACT *as shown in Figure 3(a).*

### A.4.1 ANALYSIS ON CONSECUTIVE STEPS

To bound the expectation of the global objective function at time $T$ from its optimal value, we first consider to analyze the global weight from the optimal weights by calculating single step SGD:

$$\begin{aligned}
\|\bar{\mathbf{w}}_{t+1} - \mathbf{w}^*\|^2 &= \|\bar{\mathbf{w}}_t - \eta_t g_t - \mathbf{w}^* - \eta_t \bar{g}_t + \eta_t \bar{g}_t\|^2 \\
&= \|\bar{\mathbf{w}}_t - \mathbf{w}^* - \eta_t \bar{g}_t\|^2 + 2\eta_t \langle \bar{\mathbf{w}}_t - \mathbf{w}^* - \eta_t \bar{g}_t, \bar{g}_t - g_t \rangle + \eta_t^2 \|\bar{g}_t - g_t\|^2.
\end{aligned} \tag{3}$$

The first term of Equation. 3 can be expressed as

$$\|\bar{\mathbf{w}}_t - \mathbf{w}^* - \eta_t \bar{g}_t\|^2 = \|\bar{\mathbf{w}}_t - \mathbf{w}^*\|^2 - 2\eta_t \langle \bar{\mathbf{w}}_t - \mathbf{w}^*, \bar{g}_t \rangle + \eta_t^2 \|\bar{g}_t\|^2. \tag{4}$$

The second term of Equation. 4 can be expressed as

$$\begin{aligned}
-2\eta_t \langle \bar{\mathbf{w}}_t - \mathbf{w}^*, \bar{g}_t \rangle &= -2\eta_t \sum_{k=1}^m p_{k,t} \langle \bar{\mathbf{w}}_t - \mathbf{w}^*, \nabla LM_k(\mathbf{w}_t^k) \rangle \\
&= -2\eta_t \sum_{k=1}^m p_{k,t} \langle \bar{\mathbf{w}}_t - \mathbf{w}_t^k, \nabla LM_k(\mathbf{w}_t^k) \rangle \\
&\quad -2\eta_t \sum_{k=1}^m p_{k,t} \langle \mathbf{w}_t^k - \mathbf{w}^*, \nabla LM_k(\mathbf{w}_t^k) \rangle.
\end{aligned} \tag{5}$$

By Cauchy-Schwarz inequality and AM-GM inequality, we have

$$-2\eta_t \sum_{k=1}^m p_{k,t} \langle \bar{\mathbf{w}}_t - \mathbf{w}_t^k, \nabla LM_k(\mathbf{w}_t^k) \rangle \leq \frac{1}{\eta_t} \|\bar{\mathbf{w}}_t - \mathbf{w}_t^k\|^2 + \eta_t \|\nabla LM_k(\mathbf{w}_t^k)\|^2. \tag{6}$$

By the $\mu$-strong convexity of $LM_k(\cdot)$, we have

$$-2\eta_t \sum_{k=1}^m p_{k,t} \langle \mathbf{w}_t^k - \mathbf{w}^*, \nabla LM_k(\mathbf{w}_t^k) \rangle \leq -(LM_k(\mathbf{w}_t^k) - LM_k(\mathbf{w}^*)) - \frac{\mu}{2}\|\mathbf{w}_t^k - \mathbf{w}^*\|^2. \tag{7}$$

By the convexity of $\|\cdot\|$ and the L-smoothness of $LM_k(\cdot)$, we can express third term of Equation. 4 as

$$
\begin{aligned}
\eta_t^2\|\bar{g}_t\|^2 &\leq \eta_t^2 \sum_{k=1}^m p_{k,t}\|\nabla LM_k(\mathbf{w}_t^k)\|^2 \\
&\leq 2L\eta_t^2 \sum_{k=1}^m p_{k,t}(LM_k(\mathbf{w}_t^k) - LM_k^*).
\end{aligned}
\tag{8}
$$

Combining Equations. $4 - 8$, we have

$$
\begin{aligned}
\|\bar{\mathbf{w}}_t - \mathbf{w}^* - \eta_t \bar{g}_t\|^2 \leq & \|\bar{\mathbf{w}}_t - \mathbf{w}^*\|^2 + 2L\eta_t^2 \sum_{k=1}^m p_{k,t}(LM_k(\mathbf{w}_t^k) - LM_k^*) \\
& + \eta_t \sum_{k=1}^m p_{k,t}(\frac{1}{\eta_t}\|\bar{\mathbf{w}}_t - \mathbf{w}_t^k\|^2 + \eta_t\|\nabla LM_k(w_t^k)\|^2) \\
& - 2\eta_t \sum_{k=1}^m p_{k,t}((LM_k(\mathbf{w}_t^k) - LM_k(\mathbf{w}^*)) + \frac{\mu}{2}\|\mathbf{w}_t^k - \mathbf{w}^*\|^2) \\
= & (1 - \mu\eta_t)\|\bar{\mathbf{w}}_t - \mathbf{w}^*\|^2 + \sum_{k=1}^m p_{k,t}\|\bar{\mathbf{w}}_t - \mathbf{w}_t^k\|^2 \\
& + 2L\eta_t^2 \sum_{k=1}^m p_{k,t}(LM_k(\mathbf{w}_t^k) - LM_k^*) + \eta_t^2 \sum_{k=1}^m p_{k,t}\|\nabla LM_k(\mathbf{w}_t^k)\|^2 \\
& - 2\eta_t \sum_{k=1}^m p_{k,t}(LM_k(\mathbf{w}_t^k) - LM_k(\mathbf{w}^*)) \\
\leq & (1 - \mu\eta_t)\|\bar{\mathbf{w}}_t - \mathbf{w}^*\|^2 + \sum_{k=1}^m p_{k,t}\|\bar{\mathbf{w}}_t - \mathbf{w}_t^k\|^2 \\
& + 4L\eta_t^2 \sum_{k=1}^m p_{k,t}(LM_k(\mathbf{w}_t^k) - LM_k^*) \\
& - 2\eta_t \sum_{k=1}^m p_{k,t}(LM_k(\mathbf{w}_t^k) - LM_k(\mathbf{w}^*)),
\end{aligned}
\tag{9}
$$

where we use the L-smoothness of $LM_k(\cdot)$ in the last inequality.

We use $\gamma_t = 2\eta_t(1 - 2L\eta_t)$, and the last two terms of Equation. 9 are

$$
4L\eta_t^2 \sum_{k=1}^m p_{k,t}(LM_k(\mathbf{w}_t^k) - LM_k^*) - 2\eta_t \sum_{k=1}^m p_{k,t}(LM_k(\mathbf{w}_t^k) - LM_k(\mathbf{w}^*))
$$

$$
= -\gamma_t \sum_{k=1}^m p_{k,t}(LM_k(\mathbf{w}_t^k) - GM^*) - \gamma_t \sum_{k=1}^m p_{k,t}(GM^* - LM_k^*)
$$

$$
+ 2\eta_t \sum_{k=1}^m p_{k,t}(LM_k(\mathbf{w}^*) - LM_k^*)
$$

$$
= -\gamma_t \sum_{k=1}^m p_{k,t}(LM_k(\mathbf{w}_t^k) - GM^*) - \gamma_t \sum_{k=1}^m p_{k,t}(GM^* - LM_k^*) \tag{10}
$$

$$
+ 2\eta_t \sum_{k=1}^m p_{k,t}(GM^* - LM_k^*)
$$

$$
= -\gamma_t \sum_{k=1}^m p_{k,t}(LM_k(\mathbf{w}_t^k) - GM^*) + (2\eta_t - \gamma_t) \sum_{k=1}^m p_{k,t}(GM^* - LM_k^*)
$$

$$
= -\gamma_t \sum_{k=1}^m p_{k,t}(LM_k(\mathbf{w}_t^k) - GM^*) + 4L\eta_t^2\Gamma,
$$

where $\Gamma = \sum_{k=1}^m p_{k,t}(GM^* - LM_k^*) = GM^* - \sum_{k=1}^m p_{k,t}LM_k^*$.

The first term of Equation. 10

$$
\sum_{k=1}^m p_{k,t}(LM_k(\mathbf{w}_t^k) - GM^*)
$$

$$
= \sum_{k=1}^m p_{k,t}(LM_k(\mathbf{w}_t^k) - LM_k(\bar{\mathbf{w}}_t)) + \sum_{k=1}^m p_{k,t}(LM_k(\bar{\mathbf{w}}_t) - GM^*)
$$

$$
\geq \sum_{k=1}^m p_{k,t}\langle \nabla LM_k(\bar{\mathbf{w}}_t), \mathbf{w}_t^k - \bar{\mathbf{w}}_t \rangle + \sum_{k=1}^m p_{k,t}(LM_k(\bar{\mathbf{w}}_t) - GM^*)
$$

$$
= \sum_{k=1}^m p_{k,t}\langle \nabla LM_k(\bar{\mathbf{w}}_t), \mathbf{w}_t^k - \bar{\mathbf{w}}_t \rangle + GM(\bar{\mathbf{w}}_t) - GM^* \tag{11}
$$

$$
\geq -\frac{1}{2} \sum_{k=1}^m p_{k,t}(\eta_t \|LM_k(\bar{\mathbf{w}}_t)\|^2 + \frac{1}{\eta_t}\|\mathbf{w}_t^k - \bar{\mathbf{w}}_t\|^2) + GM(\bar{\mathbf{w}}_t) - GM^*
$$

$$
\geq -\sum_{k=1}^m p_{k,t}(\eta_t L(LM_k(\bar{\mathbf{w}}_t) - LM_k^*) + \frac{1}{2\eta_t}\|\mathbf{w}_t^k - \bar{\mathbf{w}}_t\|^2) + GM(\bar{\mathbf{w}}_t) - GM^*,
$$

where the first inequality results from the convexity of $LM_k(\cdot)$, the second inequality from AM-GM inequality and the third inequality from L-smoothness of $LM_k(\cdot)$.

Therefore, Equation. 10 becomes

$$
\begin{aligned}
&-\gamma_t \sum_{k=1}^{m} p_{k,t}(LM_k(\mathbf{w}_t^k) - GM^*) + 4L\eta_t^2\Gamma \\
\leq&\gamma_t(\sum_{k=1}^{m} p_{k,t}(\eta_t L(LM_k(\bar{\mathbf{w}}_t) - LM_k^*) + \frac{1}{2\eta_t}\|\mathbf{w}_t^k - \bar{\mathbf{w}}_t\|^2)) \\
&-\gamma_t(GM(\bar{\mathbf{w}}_t) - GM^*) + 4L\eta_t^2\Gamma \\
=&\gamma_t(\sum_{k=1}^{m} p_{k,t}(\eta_t L(LM_k(\bar{\mathbf{w}}_t) - GM^*) + \frac{1}{2\eta_t}\|\mathbf{w}_t^k - \bar{\mathbf{w}}_t\|^2)) \\
&+ \gamma_t\eta_t L\Gamma - \gamma_t(GM(\bar{\mathbf{w}}_t) - GM^*) + 4L\eta_t^2\Gamma \\
=&\gamma_t(\eta_t L - 1)\sum_{k=1}^{m} p_{k,t}(LM_k(\bar{\mathbf{w}}_t) - GM^*) \\
&+ \frac{\gamma_t}{2\eta_t}\sum_{k=1}^{m} p_{k,t}\|\mathbf{w}_t^k - \bar{\mathbf{w}}_t\|^2 + (4L\eta_t^2 + \gamma_t\eta_t L)\Gamma,
\end{aligned}
\tag{12}
$$

With $GM(\bar{\mathbf{w}}_t) - GM^* > 0$ and $\eta_t L - 1 < 0$, we have

$$
\gamma_t(\eta_t L - 1)\sum_{k=1}^{m} p_{k,t}(LM_k(\bar{\mathbf{w}}_t) - GM^*) \leq 0,
\tag{13}
$$

and recall $\gamma_t = 2\eta_t(1 - 2L\eta_t)$, so $\frac{\gamma_t}{2\eta_t} \leq 1$ and $4L\eta_t^2 + \gamma_t\eta_t L \leq 6L\eta_t^2$.
Therefore,

$$
-\gamma_t \sum_{k=1}^{m} p_{k,t}(LM_k(\mathbf{w}_t^k) - GM^*) + 4L\eta_t^2\Gamma \leq \sum_{k=1}^{m} p_{k,t}\|\mathbf{w}_t^k - \bar{\mathbf{w}}_t\|^2 + 6L\eta_t^2\Gamma.
\tag{14}
$$

Thus, Equation. 9 becomes

$$
\|\bar{\mathbf{w}}_t - \mathbf{w}^* - \eta_t\bar{g}_t\|^2 \leq (1 - \mu\eta_t)\|\bar{\mathbf{w}}_t - \mathbf{w}^*\|^2 + 2\sum_{k=1}^{m} p_{k,t}\|\mathbf{w}_t^k - \bar{\mathbf{w}}_t\|^2 + 6L\eta_t^2\Gamma.
\tag{15}
$$

## A.5 BOUND FOR VARIANCE OF GRADIENTS

Next, to bound the gradient, using assumption 3, we have

$$
\begin{aligned}
\mathbb{E}\|g_t - \bar{g}_t\|^2 &= \mathbb{E}\|\sum_{k=1}^{m} p_{k,t}(\nabla LM_k(\mathbf{w}_t^k, \xi_t^k) - \nabla LM_k(\mathbf{w}_t^k))\|^2 \\
&= \sum_{k=1}^{m} p_{k,t}^2\mathbb{E}\|\nabla LM_k(\mathbf{w}_t^k, \xi_t^k) - \nabla LM_k(\mathbf{w}_t^k)\|^2 \\
&\leq \sum_{k=1}^{m} p_{k,t}^2\sigma_k^2.
\end{aligned}
\tag{16}
$$

### A.5.1 BOUND FOR DIVERGENCE OF WEIGHTS

Based on Assumption 5, for malicious clients $k = 1, 2, \ldots, c$, we have

$$
\begin{aligned}
p_{k,t} &= softmax(RS_{km}^t) \\
&= \frac{e^{RS_{km}^t}}{\sum_{i=1}^m RS_i^t} \\
&= \frac{e^{RS_{km}^{t-M} - \delta_m}}{\sum_{i=1}^c e^{RS_i^{t-M} - \delta_m} + \sum_{i=c+1}^m e^{RS_i^{t-M} + \delta_b}} \\
&= \frac{e^{RS_{km}^{t-M}}}{\sum_{i=1}^c e^{RS_i^{t-M}} + \sum_{i=c+1}^m e^{RS_i^{t-M} + \delta_b + \delta_m}} \\
&\leq \frac{e^{RS_{km}^{t-M}}}{\sum_{i=1}^c e^{RS_i^{t-M}} + \sum_{i=c+1}^m e^{RS_i^{t-M}}} \\
&= p_{k,t-M}.
\end{aligned}
\tag{17}
$$

To bound the weights, we assume within $E$ communication steps, there exists $t_0 < t$, such that $t - t_0 \leq E - 1$ and $\mathbf{w}_{t_0}^k = \bar{\mathbf{w}}_{t_0}$ for all $k = 1, 2, \ldots, m$. And we know $\eta_t$ is non-increasing and $\eta_{t_0} \leq 2\eta_t$. With the fact $\mathbb{E}\|X - \mathbb{E}X\|^2 \leq \mathbb{E}\|X\|^2$ and Jensen inequality, we have

$$
\begin{aligned}
\mathbb{E}\sum_{k=1}^m p_{k,t}\|\bar{\mathbf{w}}_t - \mathbf{w}_t^k\|^2 &\leq \mathbb{E}\sum_{k=1}^m p_{k,t}\|\bar{\mathbf{w}}_{t_0} - \mathbf{w}_t^k\|^2 \\
&\leq \sum_{k=1}^m p_{k,t}\mathbb{E}\sum_{t_0}^{t-1}(E-1)\eta_t^2\|LM_k(\mathbf{w}_t^k, \xi_t^k)\|^2 \\
&\leq \sum_{k=1}^m p_{k,t}\mathbb{E}\sum_{t_0}^{t-1}(E-1)\eta_{t_0}^2 G^2 \\
&\leq \sum_{k=1}^m p_{k,t}\mathbb{E}(E-1)^2\eta_{t_0}^2 G^2 \\
&= \sum_{k=1}^c p_{k,t}\mathbb{E}(E-1)^2\eta_{t_0}^2 G^2 + \sum_{k=c+1}^m p_{k,t}\mathbb{E}(E-1)^2\eta_{t_0}^2 G^2 \\
&\leq \sum_{k=1}^c p_{k,t}\mathbb{E}(E-1)^2\eta_{t_0}^2 G^2 + 4\eta_t^2(E-1)^2 G^2 \\
&\leq 4\sum_{k=1}^c p_{k,t}\eta_t^2(E-1)^2 G^2 + 4\eta_t^2(E-1)^2 G^2 \\
&\leq 4\sum_{k=1}^c p_{k,0}\eta_t^2(E-1)^2 G^2 + 4\eta_t^2(E-1)^2 G^2,
\end{aligned}
\tag{18}
$$

where $p_{k,0}$ is the initial probability of $k$th malicious client.

### A.5.2 CONVERGENCE BOUND

Combining Equation.(3)(15)(16)(18), we have

$$
\begin{aligned}
\|\bar{\mathbf{w}}_{t+1} - \mathbf{w}^*\|^2 &= \|\bar{\mathbf{w}}_t - \mathbf{w}^* - \eta_t \bar{g}_t\|^2 + 2\eta_t\langle\bar{\mathbf{w}}_t - \mathbf{w}^* - \eta_t\bar{g}_t, \bar{g}_t - g_t\rangle + \eta_t^2\|\bar{g}_t - g_t\|^2 \\
&\leq (1 - \mu\eta_t)\|\bar{\mathbf{w}}_t - \mathbf{w}^*\|^2 + 2\sum_{k=1}^m p_{k,t}\|\mathbf{w}_t^k - \bar{\mathbf{w}}_t\|^2 + 6L\eta_t^2\Gamma \\
&\quad + 2\eta_t\langle\bar{\mathbf{w}}_t - \mathbf{w}^* - \eta_t\bar{g}_t, \bar{g}_t - g_t\rangle + \eta_t^2\|\bar{g}_t - g_t\|^2.
\end{aligned}
\tag{19}
$$

Since $\mathbb{E}[g_t] = \bar{g}_t$, Therefore,

$$
\begin{aligned}
\mathbb{E}\|\bar{\mathbf{w}}_{t+1} - \mathbf{w}^*\|^2 \leq & (1 - \mu\eta_t)\mathbb{E}\|\bar{\mathbf{w}}_t - \mathbf{w}^*\|^2 + 2\mathbb{E}\sum_{k=1}^{m} p_{k,t}\|\mathbf{w}_t^k - \bar{\mathbf{w}}_t\|^2 + 6L\eta_t^2\Gamma \\
& + 2\eta_t\mathbb{E}\langle \bar{\mathbf{w}}_t - \mathbf{w}^* - \eta_t\bar{g}_t, \bar{g}_t - g_t \rangle + \mathbb{E}\eta_t^2\|\bar{g}_t - g_t\|^2 \\
\leq & (1 - \mu\eta_t)\mathbb{E}\|\bar{\mathbf{w}}_t - \mathbf{w}^*\|^2 + 8\sum_{k=1}^{c} p_{k,0}\eta_t^2(E-1)^2G^2 + 8\eta_t^2(E-1)^2G^2 \\
& + 6L\eta_t^2\Gamma + \eta_t^2\sum_{k=1}^{m} p_{k,t}^2\sigma_k^2 \\
= & (1 - \mu\eta_t)\mathbb{E}\|\bar{\mathbf{w}}_t - \mathbf{w}^*\|^2 \\
& + \eta_t^2[8\sum_{k=1}^{c} p_{k,0}(E-1)^2G^2 + 8(E-1)^2G^2 + 6L\Gamma + \sum_{k=1}^{m} p_{k,t}^2\sigma_k^2]
\end{aligned}
\tag{20}
$$

We set $\eta_t = \frac{\beta}{t+\gamma}$ for some $\beta > \frac{1}{\mu}$ and $\gamma > 0$, such that $\eta_1 \leq min\{\frac{1}{\mu}, \frac{1}{4L}\} = \frac{1}{4L}$ and $\eta_t \leq 2\eta_{t+E}$. We want to prove $\mathbb{E}\|\bar{\mathbf{w}}_t - \mathbf{w}^*\|^2 \leq \frac{v}{\gamma+t}$, where $v = max\{\frac{\beta^2 B}{\beta\mu-1}, (\gamma+1)\mathbb{E}\|\bar{\mathbf{w}}_1 - \mathbf{w}^*\|^2\}$ and $B = 8\sum_{k=1}^{c} p_{k,0}(E-1)^2G^2 + 8(E-1)^2G^2 + 6L\Gamma + \sum_{k=1}^{m} p_{k,t}^2\sigma_k^2$.

Firstly, the definition of $v$ ensures that $\mathbb{E}\|\bar{\mathbf{w}}_1 - \mathbf{w}^*\|^2 \leq \frac{v}{\gamma+1}$. Assume the conclusion holds for some $t$, we have

$$
\begin{aligned}
\mathbb{E}\|\bar{\mathbf{w}}_{t+1} - \mathbf{w}^*\|^2 \leq & (1 - \mu\eta_t)\mathbb{E}\|\bar{\mathbf{w}}_t - \mathbf{w}^*\|^2 + \eta_t^2 B \\
\leq & (1 - \frac{\beta\mu}{t+\gamma})\frac{v}{t+\gamma} + \frac{\beta^2 B}{(t+\gamma)^2} \\
= & \frac{t+\gamma-1}{(t+\gamma)^2}v + [\frac{\beta^2 B}{(t+\gamma)^2} - \frac{\beta\mu-1}{(t+\gamma)^2}v] \\
\leq & \frac{v}{t+\gamma+1}.
\end{aligned}
\tag{21}
$$

By the $L$-smoothness of $GM(\cdot)$, $\mathbb{E}[GM(\bar{\mathbf{w}}_t)] - GM^* \leq \frac{L}{2}\mathbb{E}\|\bar{\mathbf{w}}_t - \mathbf{w}^*\|^2 \leq \frac{L}{2}\frac{v}{\gamma+t}$.

Thus we have

$$
\mathbb{E}[GM(\mathbf{w}_T)] - GM^* \leq \frac{2}{\mu^2} \cdot \frac{L}{\gamma+T}(\sum_{k=1}^{m} p_k^2\sigma_k^2 + 6L\Gamma + 8G^2 + 8G^2\sum_{k=1}^{c} p_{k,0} + \frac{\mu^2}{4}\|w_0 - w^*\|^2).
$$

## A.6 ADDITIONAL COMMENTS

### A.6.1 CONCRETE USE-CASE

A typical use-case would be a cross-device or a cross-silo setting where a critical model is being learned, such as, mapping to a battlefield scenario by integrating sensor data from multiple sensors or a smart farms scenario to integrate data from multiple sensor nodes.

In a sensitive setting, there could be malicious intent to degrade the training process. Here an attacker could either compromise participating devices or join the training under multiple aliases. Having joined the network, it could intercept the communication between benign clients and the server, or directly compromise the server to get access to the benign gradients in order to craft the malicious gradients. This constitutes a full-knowledge attack. Further, prior work has shown that even the partial knowledge attack can be quite damaging (in such an attack, the adversary does not need to know the gradients of the benign clients, but assumes its own before-attack gradients to be an approximate estimate of the benign gradients). With an approximate estimate of the benign gradients, it is easy for an adversary to launch the directed-deviation attack to start pushing the global model away from the optima until the goal of "denial-of-service" is achieved.

### A.6.2 GENERALIZABILITY OF THE DEFENSE

We have demonstrated that TESSERACT achieves the goal of secure aggregation in the presence of directed-deviation attacks, their adaptive-white-box versions, as well targeted and untargeted label-flipping attacks. This is because the fundamental intuition behind TESSERACT is based on the assumption that a large flip-score is indicative of an attack. It should be noted that the primary objective of any untargeted model poisoning attack is to push the global model away from the optima. Given the currently aggregated global model is benign, an attack will invariably push the model parameters in the opposite direction as the benign gradients, and would consequently generate a large flip-score and get flagged. If the current global model itself had a poisoned iteration, the attacked gradients will support the current direction in which the model was moving, and would generate low flip-scores while the benign gradients now would generate high values of flip-score, as demonstrated in Figure 2. Since, we trim out both the ends of the flip-score spectrum in a relative manner, we provide a mechanism to defend against *any* untargeted attack.

### A.6.3 ACCURACY OF BASELINE FEDSGD MODEL

As in other federated learning-related security papers Fang et al. (2020); Fung et al. (2020); Bagdasaryan et al. (2020), our goal was not necessarily to achieve the smallest error rates for the protocols on the considered datasets as our goal was not to search for the most optimized DNN architecture. Instead, our goal was to demonstrate that the state-of-the-art attacks can increase the testing error rates of the learned DNN classifiers and TESSERACT can reverse the attacks better than the other defense algorithms. Since all baselines and our solution are evaluated on top of the same trained model, therefore we have a valid comparison base. Of course, we want DNNs to be reasonably performant as ours are.

