# OpenReview forum: "Tesseract: Gradient Flip Score to Secure Federated Learning against Model Poisoning Attacks"
_ICLR.cc/2022/Conference — ICLR 2022 Submitted_

### Official Review · Reviewer_VCnh · 2021-10-30

**Correctness:** 3
**Technical Novelty And Significance:** 3
**Empirical Novelty And Significance:** 3
**Recommendation:** 8
**Confidence:** 4

**Main Review:**

I enjoyed reading the paper. In contrast to many research efforts on adversarial ML, this paper makes many security assumptions that set it apart with respect to the existing body of literature. I also praise the the consideration of attackers with varying “strength” and the different datasets. All these points make me lean to recommend acceptance.

Nonetheless, there are some issues that the authors could solve to further improve their paper. Let me elaborate on the above-mentioned weaknesses, starting from the most significant ones.

**Assumptions and Threat Model?**
This is probably the only “true” problem of the paper, which should be absolutely rectified.
I was not fully able to understand the assumptions made by Tesseract. Does it work “only” against the “directed deviation attack” proposed by Fang et al.? Or does it also protect against different attacks?
In general, Section 2.2, Threat Model, is not very comprehensive. The authors should better expand this section by clearly pointing out all the assumptions and requirements of the proposed method. This is especially true because the Fang et al. attack was proposed in 2020, and some of its assumptions are not yet well-known. Specifically, this statement is suspicious: “We assume a full-knowledge (white-box)
attack where the attackers have access to the current benign gradients.”. Does it mean that Tesseract only works under this assumption? I.e., the attacker knows, and exploits, the current benign gradients? This is a rather “unrealistic” assumption: I understand the willingness to work against “worst case” scenarios; yet, if such “worst case” scenarios are not realistic in the first place, then what is the purpose of the proposed mechanism? What benefit is there in protecting against an attack that will never happen in the first place?
I invite the authors to restructure this section by using the common taxonomies adopted in adversarial ML papers [I].


**Problem or Feature Space attacks?**
The authors perform their experiments on four well-known datasets: MNIST, CIFAR, Shakespeare, FEMNIST; for each dataset, a different (deep) ML model is targeted. Three of these datasets are of images, whereas Shakespeare contains text data.
There are different ways to create “adversarial examples”, depending on the ‘space’ where the perturbation is applied. As far as I am aware, the adversarial examples considered in this paper to perform the poisoned updates are created in the feature space. It would be a lot more interesting if at least one evaluation included adversarial examples generated in the “problem” space [A]—or, at the very least, considered samples generated by “physically realizable” adversarial perturbations [B]. I acknowledge that the method should work even in these circumstances, as the proposed Tesseract defense is agnostic of the process used to apply the perturbation. However, considering the strong relationship with (real) security that permeates the paper, I believe that a more convincing use-case would dramatically improve the quality of the paper. This is also motivated by the current state-of-the-art: after almost a decade of adversarial attacks, more recent efforts are leaning towards evaluation that consider more realistic circumstances, where the attacker is constrained by the limitations of the real world; this is even more true in “distributed system” scenarios, such as Network Intrusion Detection Systems, which bear a strong relationship with federated learning (e.g., [C, D, E, F]).
As such, I invite the authors to perform an additional “proof-of-concept” experiment where they consider adversaries with constrained capabilities. This is also motivated by the fact that some perturbations may yield different effects when created in the problem space (as shown in [A]).

**Tradeoff?**
A common problem in adversarial ML countermeasures is that they may degrade baseline performance [G, H]. Hence, I am interested in knowing how the proposed method responds when there are no “malicious” clients. Even if the baseline performance does not decrease, what is the overhead of the proposed method? For instance, in Table 2 the authors report some results for “Attack=None”, which I assume represent the accuracy when no attack takes place. However, all the rows of these experiments (namely, FedSGD, Tesseract, Faba, FoolsGold, FLTrust) consider hardening FL techniques; for instance, on MNIST the proposed Tesseract has an accuracy of 92.52 when no attack takes place—the best among all other defences. Despite being appreciable, I am interested in knowing the performance when NO defense is applied. Surely, the test accuracy in a “fully trusted” FL setting should be superior than 92.52. Hence, I ask: what is the ‘cost’ of Tesseract?

**Lack of a concrete use-case.**
I believe that the paper could be further improved with a concrete use-case, where the authors explain, step-by-step, how a (single, or multiple) attacker can compromise a federated learning system, and how the proposed method can help in solving such problem. Hence, I request the description of a concrete use-case explaining the abstract scenario reported in Figure 1. Such use-case can be at the basis of the “constrained” attack that I invite the authors to perform in my "problem space perturbations" suggestion.





Some additional issues:

•	In the Introduction, the authors state: “To counter this threat, a set of approaches has been developed for countering Byzantine clients in FL…”. I believe that “Byzantine Clients” is a wrong term: what is countered by Tesseract are not byzantine clients, but "unloyal" clients, that are “against” the byzantine clients (at least by referring to the well-known problem of the byzantine generals, which should agree on a method to reach consensus in the presence of unloyal generals).

•	The caption of Figure 1 has a typo “c out of m clients maybe be malicious”.

•	In Figure, the gradient “LM_{c-1}” is out of place.

•	In Section 2, the authors state “Our simulation of federated learning consists of m clients, each with its own local data, but the same model architecture and SGD optimizer, out of which c are malicious, as shown in Figure 1”. Is there a minimum amount of “m”?

•	Figure 1 appears before Figure 2, but in the text it is referenced after Figure 2.

•	Putting Figure 2 so early on is very confusing. The “flip score” is a measure introduced for the first time in this paper. As such, any reader would be thrown off by such graphs before reading the paper, meaning that the findings of Figure 2 are difficult to interpret---during the Introduction---, as the flip score has not been defined yet. As such, such graphs are ultimately meaningless: I have to trust the authors that they correspond to “interesting” observations and “fair” experiments, which is not scientific.

•	The presentation and notation in the “Flip-score” (page 5) is very ugly and difficult to follow.

•	Section 5 should be merged in Section 6

•	W.r.t. Table 2, the authors state “We see that TESSERACT is the winner or 2nd place finisher in 7 of the 12 cells (benign + two attacks * 4 datasets)”. This should be better highlighted. I only see three bold values for Tesseract in Table 2.

•	W.r.t. Table 2, the authors state “We have not shown the test loss curve for Krum aggregation because of the large loss values.”. I invite the authors to report such values in Table 2, because the different “formats” of the three subtables (None, Full-Krum, Full-Trim) make this table very hard to interpret.



EXTERNAL REFERENCES

[A]: "Intriguing properties of adversarial ml attacks in the problem space." 2020 IEEE Symposium on Security and Privacy (SP). IEEE, 2020.

[B]: "Improving robustness of ML classifiers against realizable evasion attacks using conserved features." 28th {USENIX} Security Symposium ({USENIX} Security 19). 2019.

[C]: "Modeling Realistic Adversarial Attacks against Network Intrusion Detection Systems." ACM Digital Threats: Research and Practice. 2021.

[D]: "Constrained concealment attacks against reconstruction-based anomaly detectors in industrial control systems." ACM Annual Computer Security Applications Conference. 2020.

[E]: "Conaml: Constrained adversarial machine learning for cyber-physical systems." Proceedings of the 2021 ACM Asia Conference on Computer and Communications Security. 2021.

[F]: "Resilient networked AC microgrids under unbounded cyber attacks." IEEE Transactions on Smart Grid 11.5 (2020): 3785-3794.

[G]: "Adversarial example defense: Ensembles of weak defenses are not strong." 11th {USENIX} workshop on offensive technologies ({WOOT} 17). 2017.

[H]: "Deep reinforcement adversarial learning against botnet evasion attacks." IEEE Transactions on Network and Service Management 17.4 (2020): 1975-1987.

[I]: "Wild patterns: Ten years after the rise of adversarial machine learning." Pattern Recognition 84 (2018): 317-331.


**Summary Of The Paper:**

The paper tackles the problem of adversarial attacks in federated learning settings. The main proposal is a defensive technique to address the “byzantine generals” problem in federated learning: how to ensure that the general ML model is not affected by “poisonous” attempts made by corrupted clients.
The proposed technique is experimentally validated on four datasets, outperforms previous defensive methods, and the evaluation also considers adaptive adversaries with increasing degrees of knowledge.

Overall, the presentation of the paper is very good.
The quality of the English text is good.
Figures are appropriate, Tables require some editing.
The topic addressed by the manuscript is trendy and in-line with ICLR’s scope.
The references should be improved
The contribution is significant

STRENGTHS:
+ Adaptive adversary
+ Trendy subject (federated learning)
+ Evaluation on multiple datasets
+ Technically sound

WEAKNESSES
- Unclear assumptions and threat model.
- Problem or Feature space attacks?
- Lack of a concrete use-case
- Tradeoff?


**Summary Of The Review:**

The paper tackles a very interesting problem and the many security considerations as well as the experiments on various datasets and comparisons with existing defenses are commendable.

Some issues (unclear threat model, flexbile perturbations in the feature space) still prevent me from recommending complete acceptance. More clarifications are necessary, and by adding some more "realistic" experiments I believe that the paper could be turned into a significant submission of ICLR.

I am recommending a "5", but my score can be easily increased to 6 by addressing the many clarifications expressed in my review. Further experiments and the concrete use-case would further increase my score.

___

AFTER REBUTTAL: score increased to 6 which I will increase additionally to 8 if the authors are willing to support the claim that Tesseract is a "secure by design" defense.

FURTHER UPDATE: score increased to 8, and I stand by my decision unless other reviewers point out that the claim of Tesseract being "secure-by-design" is flawed.

---

> ### Author Response · Authors · 2021-11-14
> **Defense assumptions, threat model, tradeoff, use-case**
>
> (1/2)
>
> We thank the reviewer for his appreciation of our submission and look forward to iterating to improve the paper further for ICLR 2022.
>
> 1) Assumptions and threat model: We agree that the threat model should have been described in a clearer way and we summarize it here for clarity. Given the intuition behind TESSERACT, it works against any attack that pushes the global model against the direction of the optima, which is the primary objective of an untargeted model poisoning attack. Any such attack would invariably generate a large flip-score (by definition), and would be flagged by our defense. We have evaluated Tesseract on the directed deviation attack [7] and label-flipping attack [1]. However, the intuition behind our design is general enough to work against other untargeted model poisoning attacks as well.
>
>     We have followed the standard practice of machine learning security where we demonstrate the white-box attack-based evaluation, that is, performance of the defense technique against the worst-case scenario. Such white-box attacks have been used to stress test the defense model in all recent competitive works in this space including [2], [3] and the seminal work that started this approach is [4]. Defense against the worst case scenarios guarantees security in situations that are less severe.
>
>     “We assume a full-knowledge (white-box) attack where the attackers have access to the current benign gradients.” This is the assumption that the attacker makes in order to craft the attack and we prove that TESSERACT still provides an accurate model under such a powerful adversary. By definition, an attacker with less knowledge or power than this white-box attacker will be able to inflict less harm on TESSERACT and the model. Thus the white-box attack is used for stress testing TESSERACT.
> We discuss the use-case of the attack in Point 4 below.
>
>     [1] -Zhang, Chiyuan, Samy Bengio, Moritz Hardt, Benjamin Recht, and Oriol Vinyals. "Understanding deep learning (still) requires rethinking generalization." Communications of the ACM 64, no. 3 (2021): 107-115.
>
>     [2] - Fung, Clement, Chris JM Yoon, and Ivan Beschastnikh. "The limitations of federated learning in sybil settings." In 23rd International Symposium on Research in Attacks, Intrusions and Defenses ({RAID} 2020), pp. 301-316. 2020.
>
>     [3] -  Cao, Xiaoyu, Minghong Fang, Jia Liu, and Neil Zhenqiang Gong. "FLTrust: Byzantine-robust Federated Learning via Trust Bootstrapping." NDSSarXiv preprint arXiv:2012.13995 (20210), pp. 1-18. 2021.
>     [4] - Athalye, Anish, Nicholas Carlini, and David Wagner. "Obfuscated gradients give a false sense of security: Circumventing defenses to adversarial examples." In International conference on machine learning, pp. 274-283. PMLR, 2018.
>
> 2) Problem or feature space attacks: We target the domain of “untargeted model poisoning attacks” in this paper that directly poison the gradients instead of crafting adversarial data samples, which is done in a “data poisoning attack”. We thus cover a wider range of attacks in the training phase, where attackers find poisoning the gradients directly to be more effective than poisoning data samples, as shown in [5] and [6], the main reason being that without access to the data of the other participating clients (the benign ones), the scope of data poisoning becomes very limited, whereas crafted gradient updates can potentially cause larger damage to the system.
>
>     [5] - Bhagoji, Arjun Nitin, Supriyo Chakraborty, Prateek Mittal, and Seraphin Calo. "Model poisoning attacks in federated learning." In In Workshop on Security in Machine Learning (SecML), collocated with the 32nd Conference on Neural Information Processing Systems (NeurIPS’18). 2018.
>
>     [6] - Fang, Minghong, Xiaoyu Cao, Jinyuan Jia, and Neil Gong. "Local model poisoning attacks to byzantine-robust federated learning." In 29th {USENIX} Security Symposium ({USENIX} Security 20), pp. 1605-1622. 2020
>
> 3) Tradeoff (Cost of secure FL through TESSERACT): In Table 2, FedSGD represents a baseline aggregation without any defense. FedSGD is basically a weighted mean aggregation of all the clients weighted by the amount of data size that a client holds.
>
>     For an estimate of the overhead, here, we report the aggregation time of competing aggregation techniques when a DNN is trained on MNIST for 20 rounds. We report the time per round averaged over the 20 rounds of aggregation.
> FedSGD - 7 ms (baseline),
> TESSERACT- 9 ms (our solution),
> Trimmed mean - 8 ms,
> FLTrust - 25 ms,
> FABA - 1 s,
> FoolsGold - 1 s
>
>     TESSERACT does not have any significant overhead as compared to the baseline FedSGD as it only involves simple vector operations to identify the sign flips and the flip-score without any need for an iterative approach as in FABA or FoolsGold. In fact, we designed the exact flip score computation of TESSERACT so that it can be done using efficient vector operations.

---

> > ### Comment · Reviewer_VCnh · 2021-11-14
> > **Worst case**
> >
> > I appreciate the authors' response, but let me go a bit further as I have still some doubts.
> >
> > I acknowledge that defensive mechanisms should be evaluated in "worst case" scenarios. However, what I am referring here is whether the proposed approach works **if, and only if,** (i) the adversary has complete knowledge (i.e., white box) of the benign gradients, and (ii) fully exploits such knowledge.
> >
> > In other words: under the assumption of a "black-box" attacker, would the proposed Tesseract method mitigate the attack, or would it result in no effect because the strategy is different (as the attacker does not have access to the benign gradient)? Please answer just with a "yes/no".

---

> > > ### Author Response · Authors · 2021-11-14
> > > **Re: Worst case**
> > >
> > > The short answer is yes our defense works against a weaker attacker as well.
> > >
> > > To provide more detail:
> > > Any attacker, to degrade the model accuracy, must send gradient updates to push the model away from the optimal. This will result in anomalous flip score, which we will detect. How effective an adversary is in pushing the model away from the optimal depends on how much knowledge he has. Our technique works for white box (full knowledge adversary), for blackbox attacks, and for weaker adversaries. In fact, all our evaluation except for Sec. 6.2, uses the blackbox adversary, i.e., one who knows what aggregation mechanism is being used but does not know the internals of TESSERACT.
> > >
> > > For completeness, if the adversary does not even know the aggregation policy, then the attack is not very effective. We have seen this by mounting the Trim attack with Krum aggregation in place and vice-versa and these are not very effective attacks. Hence, these are very easy to defend and TESSERACT is capable of this, as are prior defense techniques.

---

> > > > ### Comment · Reviewer_VCnh · 2021-11-14
> > > > **Ack**
> > > >
> > > > Thanks for the clarification. This point was obscure, because the term "white box" is used many times in the paper, both for the "knowledge of the benign gradients", as well as for modeling the attacker who "knows about Tesseract".
> > > >
> > > > I believe such security considerations to be one of the strongest points of the paper, hence I carefully invite the authors to make everything more clear.
> > > >
> > > > (I will go over the remaining points of your reply later)

---

> > > > > ### Author Response · Authors · 2021-11-14
> > > > > **Terminology clarification**
> > > > >
> > > > > This is a useful conclusion to the discussion in that in the final version we will carefully define the terms:
> > > > > * black box attack
> > > > > * partial white box attack (knows what aggregation scheme is being used, knows TESSERACT is being used, but does not know the dynamically calculated parameter values of TESSERACT)
> > > > > * fully white box and adaptive (in addition to partial above, also knows the dynamically calculated parameter values of TESSERACT and actively adapts the attack to avoid detection)
> > > > >
> > > > > We appreciate the reviewer's probing that has brought this out.

---

> > ### Comment · Reviewer_VCnh · 2021-11-15
> > **Tradeoff**
> >
> > I have looked into the paper proposing FedSGD (McMahan et al. "Communication-Efficient Learning of Deep Networks from Decentralized Data"). In their paper, in Table 2, FedSGD is able to reach 99% Accuracy on MNIST.
> >
> > However, in your paper, the accuracy reported by FedSGD (in Table 2) is only 92.5. Is this normal? Moreover, the first row of Table 2 states "Test Accuracy (%) / Test Loss", but the caption says "Accuracy"). Is such discrepancy due to a different evaluation metric (Accuracy/Loss), is it a typo, or is it an expected result? I was not able to find in the paper any justification of why the baseline model performs poorly (as I pointed out in my review). The impression is that the "baseline" model is not actually a baseline because it underperforms.
> >
> > This point is crucial and requires more clarification.

---

> > > ### Author Response · Authors · 2021-11-16
> > > **FedSGD accuracy**
> > >
> > > Good catch and we will add an explanation to the paper in the camera-ready version. As in other federated learning-related security papers [1-3], our goal was not necessarily to achieve the smallest error rates for the protocols on the considered datasets as our goal was not to search for the most optimized DNN architecture. Instead, our goal was to demonstrate that the state-of-the-art attacks can {\em increase} the testing error rates of the learned DNN classifiers and TESSERACT can reverse the attacks better than the other defense algorithms. Of course, we want DNNs to be reasonably performant as ours are and as pointed out by the reviewer.
> > >
> > > Test accuracy (%) is the metric used for all datasets, except the NLP dataset Shakespeare. For NLP dataset, as is common practice, we use the Test Loss metric. This is implied in the table heading row (though in hindsight, this could be made clearer): "Test accuracy (%) / Test loss (only for Shakespeare)".
> > >
> > > [1] Fang, Minghong, Xiaoyu Cao, Jinyuan Jia, and Neil Gong. "Local model poisoning attacks to byzantine-robust federated learning." In 29th USENIX Security Symposium (USENIX Security 20), pp. 1605-1622. 2020.
> > >
> > > [2] Fung, Clement, Chris JM Yoon, and Ivan Beschastnikh. "Mitigating sybils in federated learning poisoning." International Symposium on Research in Attacks, Intrusions and Defenses (RAID), pp. 1-15. 2018.
> > >
> > > [3] Li, Tian, Anit Kumar Sahu, Manzil Zaheer, Maziar Sanjabi, Ameet Talwalkar, and Virginia Smith. "Federated optimization in heterogeneous networks." International Conference on Machine Learning (ICML) workshop, pp. 1-6. 2019.

---

> ### Author Response · Authors · 2021-11-14
> **Continuation of response**
>
> (2/2)
>
> 3) (contd.) TESSERACT maintains the integrity of every gradient update vector and does not perturb any inputs and therefore does not impact the accuracy of training, unlike with differential privacy [7].
>
>     [7] Naseri, Mohammad, Jamie Hayes, and Emiliano De Cristofaro. "Toward robustness and privacy in federated learning: Experimenting with local and central differential privacy." 29th Network and Distributed System Security Symposium (NDSS) 2022
>
>
>
> 4) Concrete use-case: We appreciate this recommendation and we will make sure to include this when allowed to update. A typical use-case would be a cross-device setting where some critical model is being learned, such as, mapping to a battlefield scenario by integrating sensor data from multiple sensors or a smart farms scenario to integrate data from multiple sensor nodes.
>
>     In a sensitive setting, there could be a malicious intent to degrade the training process. Here an attacker could either compromise participating devices or join the training under multiple aliases. Having joined the network, it could intercept the communication between benign clients and the server, or directly compromise the server to get access to the benign gradients in order to craft the malicious gradients. This constitutes a full-knowledge attack. Further, prior work has shown that even the partial knowledge attack can be quite damaging (in such an attack, the adversary does not need to know the gradients of the benign clients, but assumes its own before-attack gradients to be an approximate estimate of the benign gradients).

---

> > ### Comment · Reviewer_VCnh · 2021-11-19
> > **Update**
> >
> > I acknowledge all your replies, but I endorse the authors to provide an "updated" version of their paper that answers (at least some of) the points I raised. According to the ICLR rules, authors can update their submissions only until November 22nd.

---

### Official Review · Reviewer_EZwJ · 2021-11-02

**Correctness:** 3
**Technical Novelty And Significance:** 2
**Empirical Novelty And Significance:** 2
**Recommendation:** 5
**Confidence:** 4

**Main Review:**

The strengths and weaknesses of this paper are summarized as follows:
Strengths:
+ The problem studied in this paper is important and needs to be solved in federated learning
+ Good writing

Weaknesses:
- Need to include more related work that is highly important
- Need more justifications about the novelty claims
- Need to add some experiments under non-IID settings
- Need to unify the attack paradigm
- Insufficient theoretical analysis

Comments:
1. The following important references are missing:
[1] Kang J, Xiong Z, Niyato D, et al. Incentive mechanism for reliable federated learning: A joint optimization approach to combining reputation and contract theory[J]. IEEE Internet of Things Journal, 2019, 6(6): 10700-10714.
[2] Awan S, Luo B, Li F. CONTRA: Defending against Poisoning Attacks in Federated Learning[C]//European Symposium on Research in Computer Security. Springer, Cham, 2021: 455-475.
[3] Zhang J, Wu Y, Pan R. Incentive Mechanism for Horizontal Federated Learning Based on Reputation and Reverse Auction[C]//Proceedings of the Web Conference 2021. 2021: 947-956.
2. The untargeted model poisoning attacks (i.e., Full-Krum attack and Full-Trim attack) designed in this paper are vague. It would be better if the authors could formally define these attacks. Second, the authors need to explain how these attacks in this paper differ from [4].
[4] Minghong Fang, Xiaoyu Cao, Jinyuan Jia, and Neil Zhenqiang Gong. Local model poisoning attacks to byzantine-robust federated learning. In 29th USENIX Security Symposium (USENIX Security 20), Boston, MA, August 2020.
3. The reputation-based techniques used in this paper are quite common and non-surprising. A lot of previous work (e.g., [1-3]) has used reputation-based techniques to mitigate the poisoning attacks. Therefore, readers may think that the authors just applied the reputation-based methods in federated learning. The authors need to provide more details to justify the novelty of this paper.
4. There are some grammatical errors and inappropriate formula symbols in the context. For example, “k=1,2,…,m” should be “$k=1,2,\ldots,m$.”
5. The editorial quality of this paper is not always satisfactory. It contains quite a lot of inconsistent/non-precise descriptions, as also reflected in the above comments.
6. Theorem 1 lacks rigorous proof and complete theoretical analysis. It would be better if the author could give complete proof of Theorem 1.
7. The number of attackers has always been a very important hyperparameter (or factor) in model poisoning attacks. Therefore, it would be better if the authors could conduct more case studies to explore the influence of the number of attackers on different defense algorithms.
8. In addition, non-IID data seems also to affect the gradient direction (or value) of the client. Therefore, the authors need to add some experiments to illustrate the effectiveness of the proposed algorithm under non-IID settings.
9. In fact, the poisoning attack defended by the baselines chosen by the authors is different from the attacks designed in this paper. Then, it would be better if the authors tested the proposed defense method on the poisoning attack involved in the baseline schemes.
10. In general, the authors need to add more theoretical analysis and verification experiments.


**Summary Of The Paper:**

This paper studied a very important topic in the field of federated learning: how to efficiently resist untargeted model poisoning attacks. In order to defend against such a poisoning attack, the authors developed TESSERACT, an aggregation algorithm that assigns reputation scores to participating clients based on their behavior in the training phase and weights the client's contribution. Extensive case studies have verified the effectiveness of the algorithm. In particular, the experimental results show that TESSERACT provides robustness against even a white-box version of the attack.

**Summary Of The Review:**

For now, the authors need to add more theoretical analysis and verification experiments. The reason is that there are still unfair comparisons in the comparative experiments in this paper. If the author can address the reviewer’s comments, I will consider giving it a score of "6".

---

> ### Author Response · Authors · 2021-11-16
> **Poisoning attacks, reputation weights, theoretical analysis, effect of number of attackers, non-IID data**
>
> (1/3)
>
> 1) Additional related work:
>
>     [1] Kang J, Xiong Z, Niyato D, et al. Incentive mechanism for reliable federated learning: A joint optimization approach to combining reputation and contract theory[J]. IEEE Internet of Things Journal, 2019, 6(6): 10700-10714.
>
>     [2] Awan S, Luo B, Li F. CONTRA: Defending against Poisoning Attacks in Federated Learning[C]//European Symposium on Research in Computer Security. Springer, Cham, 2021: 455-475.
>
>     [3] Zhang J, Wu Y, Pan R. Incentive Mechanism for Horizontal Federated Learning Based on Reputation and Reverse Auction[C]//Proceedings of the Web Conference 2021. 2021: 947-956.
>
>     Our design is based on a dynamic reputation weight computation to minimize the impact that malicious clients can have on the global model that is being iteratively computed. Reputation-based techniques have also been used in [1] and [3] to incentivize workers with high quality data to stay in the network.
>
>     [1] has designed a reputation-based worker selection scheme for federated learning using a “multiweight subjective logic”, in conjunction with blockchain to maintain untamperable and publicly available reputation scores as an incentive for workers to stay in the network.
>
>     [3] also proposes an incentive mechanism based on “reputation and reverse auction theory”.
>
>     We have made use of the reputation mechanism based on the individual flip-score of the workers with the goal to mainly identify the malicious ones and minimize their contribution to the global model.
>
>     [2] proposes a defense based on reputation scores computed from pairwise cosine similarity between workers to identify clients that have the same malicious objective. This approach is similar to that of FoolsGold [4] that we experimentally compare TESSERACT to in our submission. We have shown that such an approach breaks when the malicious clients start sending updates that differ from one another significantly, but can still poison the global model.
>
>     [4] Fung, Clement, Chris JM Yoon, and Ivan Beschastnikh. "The limitations of federated learning in sybil settings." In 23rd International Symposium on Research in Attacks, Intrusions and Defenses ({RAID} 2020), pp. 301-316. 2020.
>
> 2) Poisoning attacks: The Full-Krum and Full-Trim attacks described in our paper are from the state-of-the-art attack paper for model poisoning attacks although we show that our defense applies to other attacks such as targeted and untargeted label-flipping attack [5].  Our attack is the same as that in the paper - “Minghong Fang, Xiaoyu Cao, Jinyuan Jia, and Neil Zhenqiang Gong. Local model poisoning attacks to byzantine-robust federated learning. In 29th USENIX Security Symposium (USENIX Security 20), Boston, MA, August 2020.”
> We do not come up with new attacks, albeit, we come up with whitebox variants of the attacks (where the attacker has full knowledge of the defense’s parameters) in Section 6.2 to further stress TESSERACT’s ability to defend against them. We have also demonstrated a case-study evaluation of TESSERACT against label-flipping attack in Figure 6 in Appendix A.2.
>
>     [5] - Zhang, Chiyuan, Samy Bengio, Moritz Hardt, Benjamin Recht, and Oriol Vinyals. "Understanding deep learning (still) requires rethinking generalization." Communications of the ACM 64, no. 3 (2021): 107-115.
>
> 3) Reputation weights: The primary novelty of our work is in the design and use of “flip-score” to identify malicious behavior. Flip-score can, however, only capture the present behavior of a client. In order to learn the behavior of a client over time and to reduce the number of false positives, we have proposed a stateful model with penalty, rewards, and introduced a decay mechanism to dynamically update the reputation weights. We use these reputation weights so that the aggregation can be kept simple, that is, a weighted mean aggregation. Reputation-based techniques are common and intuitive (and have even been applied in FL setting as pointed out by the reviewer), and we have made use of the concept on top of our flip-score for added benefits.
>
>     We have carefully applied the reputation mechanism to our defense by ensuring that: occasional errors by benign clients are not penalized and there is an intuitive way of trading off how much emphasis we pay to recent behavior of clients versus farther distant behavior by setting the decay parameter.

---

> ### Author Response · Authors · 2021-11-16
> **Continued response (2/3)**
>
> (2/3)
>
> 6 ) Theoretical analysis: We have put a detailed and complete proof of Theorem 1 on the anonymized link - “https://www.dropbox.com/s/s60ve5263r63n1u/Convergence_proof_tesseract.pdf?dl=0”. This theorem shows that TESSERACT's learning converges and it provides the convergence bound.
> In short, based on Assumption 1 and 2, we show after each step of SGD, the weights of the global model get closer to the optimal weights.
> According to Assumption 3, we get the bound of gradients from each client with respect to the averaged gradients.
> Finally, by using Assumptions 4 and 5, we obtain the bound of weights from each client with respect to the global weights. With this, we complete the proof of Theorem 1.
>
> 7 )  Influence of the number of attackers on different defense algorithms: We have shown in Table 2 that almost all the defense techniques except FABA fail for one or the other dataset when $c/m=0.2$. FABA fails for $c/m=0.3$ ($c$: number of malicious clients, $m$: total number of clients), whereas TESSERACT successfully defends in all the cases shown in Table 2. Figure 3b shows that when the test accuracy with FABA falls below 20%, TESSERACT still maintains an accuracy of above 80%. It is clear that the performance of the other techniques is expected to degrade even further for higher $c/m$. We provide the evaluation of TESSERACT across higher $c/m$ ratios on MNIST dataset in Figure (a) that can be found here - “https://www.dropbox.com/s/wqh7xpxdaj0gaxj/image1-new.png?dl=0”. We can see that the test accuracy drops only for $c/m>0.45$. TESSERACT, thus provides a robust defense against a wide range of malicious nodes, much wider than any existing defense.
>
> 8 ) Evaluation on non-IID data - We have simulated a non-IID data distribution in all our experiments:
>
>     -The Shakespeare dataset was partitioned sequentially among the clients which made the distribution non-IID.
>
>     -FEMNIST from the LEAF datasets is also non-IID since it is partitioned by the users who wrote the numbers/letters.
>
>     -For the other image datasets - MNIST, FMNIST, CIFAR-10, we used a non-IID bias of 0.5 to distribute the data samples among the clients.
>
>     We have also provided the evaluation of TESSERACT across varying non-IID bias in Figure (b) uploaded on “https://www.dropbox.com/s/o4acapqkbmx6edi/image2.png?dl=0”. The results clearly show the robustness of TESSERACT in benign as well as malicious conditions for 0.1 <= bias <= 0.8 with a small dip only at the extreme case of bias = 0.9 causing the test accuracy to drop to 0.82.

---

> ### Author Response · Authors · 2021-11-17
> **Continued response (3/3)**
>
> 9) Threat model of the defense benchmarks used - We compare TESSERACT with three defense benchmarks - FoolsGold[4], FABA[6], and FLTrust[7], and with the baseline aggregation techniques of FedSGD, Trimmed Mean, Median, and Krum.
>
>     -[4] shows its evaluation against a Targeted label-flipping attack only. We show in Figure 6 in Appendix A.2 that this is a weak attack and TESSERACT defends against this attack on MNIST.
>
>     -[6] shows its evaluation against Uniform and Gaussian attacks, which have been shown by [8] to have negligible impact when Trimmed Mean aggregation is used. TESSERACT is in fact superior as a defense to Trimmed Mean as we have experimentally demonstrated.
>
>     -[7] evaluates its defense against Untargeted label-flipping attack, Full-Krum and Full-Trim attacks, and Scaling attack. We have evaluated TESSERACT against Full-Krum and Full-Trim attacks in Section 6.1, and against untargeted label-flipping attack in Appendix A.2. The results in [7] itself show that Scaling attack does not have a significant impact on Trimmed Mean whereas Full-Trim has a devastating impact on the same. We have therefore considered evaluation against a stronger attack, with even stronger adaptive modifications in our paper.
>
>     [6] - Xia, Qi, Zeyi Tao, Zijiang Hao, and Qun Li. "FABA: an algorithm for fast aggregation against byzantine attacks in distributed neural networks." In IJCAI. 2019.
>
>     [7] - Cao, Xiaoyu, Minghong Fang, Jia Liu, and Neil Zhenqiang Gong. "FLTrust: Byzantine-robust Federated Learning via Trust Bootstrapping." arXiv preprint arXiv:2012.13995 (2020).
>
>     [8] - Fang, Minghong, Xiaoyu Cao, Jinyuan Jia, and Neil Gong. "Local model poisoning attacks to byzantine-robust federated learning." In 29th USENIX Security Symposium (USENIX Security 20), pp. 1605-1622. 2020.
>
> 10) More theoretical analysis and verification experiments- This has now been done with the full proof of Theorem 1 provided and two additional experiments --- how does TESSERACT react to increasing fraction of malicious clients and to increasing non-IID bias. The experiments bear out the claims that TESSERACT is the most robust defense to untargeted model poisoning attacks in federated learning.

---

> > ### Comment · Reviewer_EZwJ · 2021-11-24
> > **Thanks for your time and responses.**
> >
> > Thank you for your detailed reply. Reviewers will revise the score based on your revisions as appropriate.

---

### Official Review · Reviewer_4KHr · 2021-11-02

**Correctness:** 3
**Technical Novelty And Significance:** 3
**Empirical Novelty And Significance:** 4
**Recommendation:** 6
**Confidence:** 4

**Main Review:**

This submission presents a reasonable and timely defense against a strong attack against data poisoning in federated learning, which is nice. The design of the defense is well-executed and the inclusion of a dynamic reputation is a good addition to its robustness. I do have a few comments,  mostly regarding the part of the paper concerning adaptive attacks:

* The submission discussses *an* adaptive attack against this defense, but I would like to understand and see more discussion on why this is a strong adaptive attack specifically. It is currently not clear to me that this is a strong adaptive attack. It seems that the attacker could mount a stronger adaptive attack by keeping track of its own reputation and send local updates are optimized under an additional linear constraint that includes its own reputation? It would be great if the authors could clarify my understanding in this matter.

* How does this defense perform compared to the other considered defenses when the number of malicious clients is unknown? For example, assuming c_max is fixed to m/5, but the actually number of malicious clients is decreased from m/5 down to 1. This would be especially interesting to compare to defense algorithms that require no knowledge of c_max.

And some minor comments and questions:
* GM and LM are used in the text before the variables are formally explained
* "We halve the reputation score of every client if any of the values grows so large that the softmax operation causes an overflow" -> I am surprised that this could even happen, given the sizes of m, n_r and mu?
* page 7: "here to how the effect on diverse datasets" ->here to show the effect on diverse datasets
* "We find empirically (result not shown) that FABA degrades fast" -> it would be better if the authors could include this result in their appendix (or clarify the sentence - I think a partial answer to this statement is contained in Fig3b?)
* The caption of Fig.2 says "(a) shows the results where only benign updates were aggregated using FEDSGD, and (b) shows the case where only malicious updates were aggregated", but the figure headings say the opposite
* For CIFAR-10 and Shakespeare only two clients are malicious, is the attack too strong to be mitigated if more malicious clients exist?

**Summary Of The Paper:**

This submission with the title "Tesseract: Gradient Flip Score to Secure Federated Learning against Model Poisoning Attacks " discusses defenses against data poisoning in federated learning. The authors propose a novel defense against the recently popularized attack "Tesseract: Gradient Flip Score to Secure Federated Learning against Model Poisoning Attacks " by Yang et al. This attack reduces model availability by sending malicious updates from compromised client that maximize sign flips in the global model gradient.
This defense then proposes a measure of change in gradient direction that can be evaluated for each local update and used to dynamically down-weight clients with a large number of flips in direction.

**Summary Of The Review:**

In summary, I think this is a decent submission. I have some questions (the central one is the strength of the adaptive attack) which I would like to discuss with the authors and I am open to changing my evaluation accordingly.

---

> ### Author Response · Authors · 2021-11-15
> **Adaptive attack, comparing defense techniques in benign conditions, clarification about Figure 2**
>
> (1/2)
>
> 1) Adaptive attack - Our adaptive white-box attack is in fact what the reviewer suggests --- the malicious nodes keep track of what gradient updates will get them flagged and therefore their reputation reduced. They then send updates that are just below that threshold. The exact reputation score is not available at the individual clients. The reviewer raises an interesting possibility, which we plan to explore in future work. This possibility is that each client (including the malicious ones) reverse engineer and calculate their reputation values from what they hear from the server. Even in that case, if a malicious node, which should have a low reputation score, tries to counteract its low reputation by sending large magnitude gradient updates, that will be flagged by TESSERACT.
>
>     We have shown that TESSERACT can successfully defend against Full-Krum and Full-Trim attacks, which are considered the most damaging model poisoning attacks against Federated Learning. Then, we give additional capabilities to the attacker that it can use in order to bypass TESSERACT. Our evaluation shows that in order to gain stealth, the attack will lose strength, as we see in Section 6.2 (on adaptive whitebox attacks). This occurs, even if the attackers collude. This attack attempts to achieve just good enough stealth that can bypass the defense by undoing the attack on only “less important parameters” to maintain the highest strength for a given level of stealth in order to bypass the defense. This still results in the adaptive attacks being less severe and not very effective against TESSERACT. Thus, an adaptive white-box attacker, with access to all the internals of TESSERACT, including dynamically determined threshold parameters, cannot bypass its defense.
>
> 2) Analysis of a case where $c \leq c_{\text{max}}$: "How does this defense perform compared to the other considered defenses when the number of malicious clients is unknown?" -> It is usually recommended to set $c_{\text{max}}$ to a conservative value so that $c \leq c_{\text{max}}$. In Table 2, we see that the training does not suffer even when $c_{\text{max}}$ is set to $m/5$ but when the actual $c$ is $0$. Defense techniques (like FoolsGold and FLTrust) that do not require setting $c_{\text{max}}$ actually filter out more clients than defense techniques that upper cap $c_{max}$ to some value (like TESSERACT and FABA), as can be seen in our analysis in Table 3. We see that in a benign case, the mean fraction of clients allowed to contribute to the global model by FoolsGold was 0.29, by FLTrust was 0.48, and by TESSERACT was 0.75. We, however, see comparable performance among all the defense techniques in a benign case reported in Table 2. In other words, TESSERACT is not hurt by having a $c_{\text{max}}$ set in a benign setting.
>
> 3) Reason to safeguard reputation score against overflow - " I am surprised that this could even happen, given the sizes of m, n_r and mu?
> " -> Let us consider a case - $m=100$, $c=20$, $\mu=1.0$. Let us analyze the extreme case here, where the same 40 clients are penalized in every iteration. The penalty in this case would be -0.6 and the reward would be +0.4, and it would accumulate in every iteration. When, say for example, we train for 2,500 iterations, 60 clients would each have a reputation score of 0.4*2500 = 1000. With such high values, we observed an overflow while converting the reputation score to reputation weights. This led us to, introduce logic to prevent the overflow as a safety mechanism.
>
> 4) Comparison with FABA - "it would be better if the authors could include this result in their appendix (or clarify the sentence - I think a partial answer to this statement is contained in Fig3b?" -> Yes, you are correct. Figure 3b shows that FABA begins to degrade even for $c/m=0.3$, as explained on Page 7. We therefore did not show the results for even higher $c/m$ for FABA because the degradation starts at 0.3. On the other hand, the degradation for TESSERACT begins at $c/m$=0.46 for the same setting with MNIST distributed among 100 clients with a non-IID bias of 0.5.

---

> > ### Comment · Reviewer_4KHr · 2021-11-19
> > **Additional Comments**
> >
> > Thank you for the extensive clarification.
> > I'd like to rephrase my question regarding point 1:
> >
> > The proposed defense can indeed defend against two recently proposed poisoning attacks and the authors verify this. However, from a security standpoint this is necessary, but not sufficient for a good defense. I am most interested in clarification why no future attack could break this defense. The two-player adversarial game investigated here allows the attacker to move second and run adaptive attacks that specifically target the defense.
> >
> > In light of this, I woud like to see additional discussion in the paper text inhowfar the proposed adaptive attack is optimal. The submission proposed adaptive-krum and adaptive-trim, but why were these attacks chosen in their current variation and why is the current variation optimal?

---

> > > ### Author Response · Authors · 2021-11-21
> > > **Evaluating TESSERACT on the proposed adaptive attack**
> > >
> > > Let us begin by describing the attacks we have implemented. Full-Krum attack finds a vector of gradients $u$ by solving an optimization problem described in [1], and every malicious client would send $u$ with an additional noise to appear different. Full-Trim attack solves a different optimization problem, as described in [1] to also come up with a vector of gradients $u$ to which every malicious client $i$ would add some noise to obtain $u_i$. The problem statement in our (two) derived versions of the above (two) attacks, namely Adaptive Trim and Adaptive Krum, is to find a set of vectors $v_i$, i=0,1,2,...,c-1 (c: #malicious clients), each with a flip-score lower than the cut-off flip-score according to the adversary’s knowledge, but summing up to the sum of vectors $u_i$ that was originally determined by the adversary. We solve this problem as described in Section 6.2. In short, we initialize $v_i$ to some target value and then undo the attack on “less important parameters” until the flip-score constraint is just met, and send the computed $v_i$ for aggregation.
> > >
> > > The reviewer suggests, if we understand correctly, to change the optimization problem to finding a set of vectors $v_i$ for clients $i$ each with a reputation weight $w_i$ such that the weighted sum of $v_i$ equals the weighted sum of $u_i$, with the flip-score constraint still being valid. We describe the detailed problem formulation in the updated paper in Appendix A.3.
> > >
> > >
> > > We ran the above simulated attack to evaluate TESSERACT on MNIST, and the resulting test accuracy plot in comparison with the baseline FedSGD with Full-Trim attack can be viewed at “ https://ibb.co/373xnc1 ”. The results have also been updated in Figure 8 in Appendix A.3. TESSERACT successfully defends against the newly proposed adaptive by reaching 90% accuracy in 500 iterations as compared to the 60% accuracy that the baseline achieves.
> > >
> > > [1] - Fang, Minghong, Xiaoyu Cao, Jinyuan Jia, and Neil Gong. "Local model poisoning attacks to byzantine-robust federated learning." In 29th {USENIX} Security Symposium ({USENIX} Security 20), pp. 1605-1622. 2020

---

> > > > ### Comment · Reviewer_4KHr · 2021-11-27
> > > > **Feedback**
> > > >
> > > > Thank you for the additional experiments. This is not quite what I had a mind, but certainly an interesting adaptive variant.
> > > >
> > > > After coming back to this submission and rereading our discussion I think I can re-phrase my main question from the previous additional comment in the following way:
> > > > "The submission shows that attacks based on gradient sign flips (krum, trim, label flipping) can be robustly defeated by the proposed defense. Are gradient sign flips a necessary mechanism of any model availability attack against federated learning systems?"

---

> > > > > ### Author Response · Authors · 2021-11-28
> > > > > **Generalizability of TESSERACT**
> > > > >
> > > > > Yes, TESSERACT gets at the root cause of any untargeted model poisoning attack, as any such attack has to perform gradient flips.
> > > > >
> > > > > Untargeted attacks with the malicious objective of rendering high test error rates in the global model need to constantly push the model away from the optima. This is done so as to prevent recovery by the benign gradient updates that are computed by the server from gradient descent optimization techniques. Therefore the malicious objective is always a gradient ascent technique, and is achieved by methods like Full-Krum and Full-Trim. Gradient ascent requires a suitably large number of adversarial clients to send gradients with directions opposite to that of the benign ones. Regardless of the method used by the attacker to implement gradient ascent, TESSERACT strikes at the root cause by blocking updates with these flipped gradients as they would invariably generate a large flip-score.

---

> > > ### Author Response · Authors · 2021-11-22
> > > **Re: Evaluating TESSERACT on the proposed adaptive attack**
> > >
> > > We have now included the precise mathematical formulation of this attack suggested by the reviewer --- we call it the "Weighted-Adaptive-Trim" attack. This has been included under Section A.3 of the Supplement.
> > >
> > > We then implemented this attack and evaluated it in the insecure environment (without our defense TESSERACT) and with our defense TESSERACT. The evaluation result is shown in Figure 8 (MNIST DNN). We find that this attack is indeed effective in the insecure environment --- it degrades the accuracy of the MNIST model from 92% to 58%. But TESSERACT is robust enough to this attack as well --- it maintains an accuracy of 90%.
> > >
> > > We appreciate the reviewer pointing out this insightful new, adaptive, white-box attack.

---

> ### Author Response · Authors · 2021-11-15
> **Clarification for Figure 2**
>
> (2/2) Contd.
>
> 5) Clarification for Figure 2 - We are sorry for the mismatched labels. We have clarified the confusion in reply to Reviewer YpRv.
>
>     For completeness, we repeat here:
> Figure (a) should have been benign updates aggregated using FEDSGD and (b) should have been malicious updates aggregated, depicting the two extreme cases in a federated learning scenario. The 2nd figure should have been labeled as “(b) Malicious gradients aggregated”. Both the experiments had benign as well as malicious clients. However, for the sake of analysis, the server identified which clients were benign or malicious, and allowed aggregation of only benign clients in Part (a) and only malicious clients in Part (b). However, as in Part (a), the malicious clients were still sending their updates to the server, which was used to compute the flip-score, but their gradients were not aggregated. This was done to show that if the global aggregation is benign, the malicious flip-score is higher than the benign flip-score. However,, but if the global aggregation itself is poisoned, the malicious score can be lower than the benign flip-score. This observation motivates us to trim both the ends of the flip-score distribution before aggregation.

---

### Official Review · Reviewer_YpRv · 2021-11-08

**Correctness:** 4
**Technical Novelty And Significance:** 2
**Empirical Novelty And Significance:** 2
**Recommendation:** 5
**Confidence:** 4

**Main Review:**

Pros:
a. The defense is based on an interesting observation that, for a sufficiently small learning rate, as the model approaches optima in the benign setting, a large number of gradients do not flip their direction with a large magnitude. And if such a behavior is observed it is indicative of an attack. The paper defines a Flip Score for every received gradient update and uses it to identify and either reward or penalize the update.

b. The history of rewards and penalties for a client is maintained as a reputation score. Normalized reputation scores are then used to compute the global model for the next round (Algorithm 1).

c. In addition to defending against the directed deviation attack, the paper also proposes two adaptive attacks (Adaptive-Krum and Adaptive-Trim) in which colluding attackers knowing the parameters of TESSERACT adjust their attack vectors to escape the cut-off flip scores.

d. The paper provides a convergence analysis sketch and compares the performance of TESSERACT with a set of Byzantine Resilient defenses (Krum, Bulyan, Trimmed Mean and Median, etc.).

Weaknesses:

1. The paper proposes a defense against a specific form of attack and does not provide any guarantees or justification about why it should generalize against other more powerful forms of model poisoning attacks. On pg. 2, the paper argues (without justification) that other attacks are less damaging and essentially weaker than the attack in Fang et. al. However, one can argue that while the directed deviation attack, is designed to prevent model convergence, there are model poisoning attacks that not only insert targeted backdoors but also allow the model to converge (Bhagoji et. al., 2019: https://arxiv.org/pdf/1811.12470.pdf, Bagdasaryan et. al. 2020: https://arxiv.org/pdf/1807.00459.pdf). These attacks are thus more powerful and can also maintain stealth.

2. While the authors provide two very interesting adaptive attacks, it is hard to generalize the strength of defense without any formal guarantees, In other words, can there be no other adaptive attack that can bypass this defense. For example, can an attacker target the reputation update mechanism over time?

3. Typically, for reasons of privacy, the gradient updates are usually encrypted before being sent to the server (secure aggregation scheme by Bonawitz et. al. 2017, https://eprint.iacr.org/2017/281.pdf). TESSERACT in its current form will be difficult to implement with such schemes.

4. In Fig. 2, both sub-figures are labeled (a) and there is an inconsistency between the sub-figure caption and the image caption. Figure caption describes (a) as an aggregation of benign updates and sub-figure caption describes (a) as an aggregation of malicious updates. Finally, the Flip Scores of the figures(y-axes) are very different, but what is confusing is that there are both benign and malicious clients in both.

**Summary Of The Paper:**

The authors propose TESSERACT, an aggregation scheme that is robust to the directed deviation attack (proposed in Fang et. al. 2020).

**Summary Of The Review:**

Please see the detailed comments above.

---

> ### Author Response · Authors · 2021-11-15
> **Attacks covered by TESSERACT, novelty claims, clarifications**
>
> 1) TESSERACT 	works against any attack that forces the global model to move in the opposite direction from that of convergence, as we have mentioned in Section 7, line 4. The directed deviation attack does that by implementing Full-Krum and Full-Trim methods that we successfully defend against. Any gradient update vector that directly opposes the direction of gradient descent, which is essentially the attribute of any untargeted model poisoning attack, is bound to have a large flip-score (by definition) and thereby flagged as suspicious by Tesseract.
>
>     On Page 2, we refer to the Gaussian attack and Label-flipping attack [1] that have been shown to have little or no impact against even the traditional aggregation techniques like Trimmed-Mean and Krum. It has been shown that these existing defenses are resilient enough to prevent such attacks [2].
>
>     In this paper, we are defending against $untargeted$ model poisoning attacks that push the global model away from the optima and thus create high classification error for all classes
>
>     [1] - Xiao, Han, Huang Xiao, and Claudia Eckert. "Adversarial label flips attack on support vector machines." In ECAI 2012, pp. 870-875. IOS Press, 2012.
>
>     [2] - Fang, Minghong, Xiaoyu Cao, Jinyuan Jia, and Neil Gong. "Local model poisoning attacks to byzantine-robust federated learning." In 29th {USENIX} Security Symposium ({USENIX} Security 20), pp. 1605-1622. 2020
>
> 2) We have presented two strong adaptive attacks that attempt to evade our defense scheme in a stealthy manner. In doing so, the attacker makes sure that it is not flagged by TESSERACT, and thus fools the server to allot it a higher reputation. Further, the reputation mechanism is in control of the server and in our threat model, the adversary compromises multiple clients and not the server. So, the adversary tries an indirect approach to act stealthy and gain in reputation, which automatically decreases the strength of the attack, as shown in Section 6.2.
>
> 3) TESSERACT requires the server to compute the flip-score for every client in order to compute its reputation weight. If gradient encryption is used such that the server is meant to decrypt the gradients (the common use of this term --- this is used to guard against curious clients and man-in-the-middle attacks), then TESSERACT works as is, since it is integrated with the aggregation server. If on the other hand, the clients use homomorphic encryption or secure MPC (because they do not completely trust the server) then TESSERACT will not be able to decrypt and access the gradients and cannot work. The paper that the reviewer points out falls in the second category and so the reviewer is correct in his statement.
>
>     Note also that gradient encryption is still a costly operation despite steady improvements in secure MPC primitives. [3] states “more than 80% of the training iteration time is spent on encryption/decryption.” and that some FL applications cannot afford encryption due to resource constraints as “encryption can inflate the amount of data size by as much as 150x” and go for less stringent privacy requirements. Thus, in a typical use case where TESSERACT is applied in a cross-device setting with many individually weak devices, such gradient encryption may not be in use
>
>     [3] - Zhang, Chengliang, Suyi Li, Junzhe Xia, Wei Wang, Feng Yan, and Yang Liu. "Batchcrypt: Efficient homomorphic encryption for cross-silo federated learning." In 2020 {USENIX} Annual Technical Conference ({USENIX}{ATC} 20), pp. 493-506. 2020.
>
> 4) We apologize for the mis-labeling of Figure 2, which is a typo. (a) should have been benign updates aggregated using FEDSGD and (b) should have been malicious updates aggregated, depicting the two extreme cases in a federated learning scenario. Both the experiments had benign as well as malicious clients. However, for the sake of analysis, the server identified which clients were benign or malicious, and allowed aggregation of only benign clients in Part (a) and only malicious clients in Part (b). However, as in Part (a), the malicious clients were still sending their updates to the server, which was used to compute the flip-score, but their gradients were not aggregated. This was done to show that if the global aggregation is robust (i.e., not fooled by the malicious clients), the flip-score of the malicious nodes is higher than that of benign nodes. However, if the global aggregation is fooled (i.e., the malicious clients succeed in pushing the global model away from the optima), then the flip-score of the malicious nodes is lower than that of the benign nodes. This observation motivates us to trim both the ends of the flip-score distribution before aggregation.

---

### Author Response · Authors · 2021-11-22
**Summary of all modifications in the revised paper**

We thank all the reviewers for the appreciation of our work, and for making the useful recommendations. We have submitted an updated version of the paper based on these comments. Below is the summary of the main additions in the paper. These are complemented by the detailed responses to the individual reviewer comments.

1. We have added the formulation of another adaptive white-box attack in response to Reviewer 2 (4KHr). This uses the knowledge of reputation scores at all adversarial clients. We perform an experiment on this attack (“Weighted-Adaptive-Trim attack”) for the MNIST model. We show that TESSERACT is resilient to this attack as well, that was specifically tailored against it while the baseline non-adaptive version of the attack, that is, Full-Trim has a devastating impact on the unprotected FL system. (Appendix A.3, Figure 8).

2. We have added micro-experiments evaluating the robustness of TESSERACT in the presence of varying number of malicious clients, and with data distributed among clients in varying degrees of non-IIDness, as a response to Reviewer EZwJ (Appendix A.3, Figure 7). The results show that the global model remains secure for a wide range of conditions - upto $c/m=0.45$ (malicious client ratio) and through a non-IID bias of 0.1 to 0.8.

3. We have included the detailed proof of our theoretical result (referenced in Section 4) in Appendix A.4 and A.5 as requested by Reviewer EZwJ. A.4 proves that TESSERACT converges, and A.5 computes the convergence bound.

4. We have added the inspiration for a reputation-based technique in Section 3 (response to Reviewer EZwJ), and emphasized upon the threat model that TESSERACT can defend against, in Sections 1 and 2 (response to Reviewer VCnh). We have also made all the editorial changes in the paper as suggested by the reviewers.

---

> ### Comment · Reviewer_VCnh · 2021-11-23
> **Odd formatting**
>
> I acknowledge the clarifications on the threat model, which is now more clear.
>
> Still, I am unable to find an explanation for the "mismatch" in the performance of FedSGD, as well as a description of a "concrete use case".
>
> Regardless, the updated version of the paper has an odd formatting. Page 10 only has a single section (Section 8) of 4 lines. Page 15 has too much spacing. I am not seeing any Appendix: currently, all the paper has 21 pages. I am questioning the adherence of the paper to ICLR's submissions rules.
>
> (for reference, I am talking about the file accessible from this link: https://openreview.net/pdf?id=XIZaWGCPl0b )

---

> > ### Author Response · Authors · 2021-11-23
> > **Mismatch in performance of FedSGSD and concrete use case**
> >
> > The explanation is there in our responses --- to the first point in the response titled “FedSGD accuracy” and to the second point in the response item titled “Concrete use-case”. For completeness we are repeating those here. We plan to include them in the main body of the paper but the page limit of 9 pages means we will have to remove something else. We are waiting on the end of the discussion period to determine what material can be removed because it is not deemed important by any of the parties.
> >
> > **FedSGD accuracy**
> >
> > “Good catch and we will add an explanation to the paper in the camera-ready version. As in other federated learning-related security papers [1-3], our goal was not necessarily to achieve the smallest error rates for the protocols on the considered datasets as our goal was not to search for the most optimized DNN architecture. Instead, our goal was to demonstrate that the state-of-the-art attacks can {\em increase} the testing error rates of the learned DNN classifiers and TESSERACT can reverse the attacks better than the other defense algorithms. Of course, we want DNNs to be reasonably performant as ours are and as pointed out by the reviewer.
> > Test accuracy (%) is the metric used for all datasets, except the NLP dataset Shakespeare. For NLP dataset, as is common practice, we use the Test Loss metric. This is implied in the table heading row (though in hindsight, this could be made clearer): "Test accuracy (%) / Test loss (only for Shakespeare)".
> > [1] Fang, Minghong, Xiaoyu Cao, Jinyuan Jia, and Neil Gong. "Local model poisoning attacks to byzantine-robust federated learning." In 29th USENIX Security Symposium (USENIX Security 20), pp. 1605-1622. 2020.
> > [2] Fung, Clement, Chris JM Yoon, and Ivan Beschastnikh. "Mitigating sybils in federated learning poisoning." International Symposium on Research in Attacks, Intrusions and Defenses (RAID), pp. 1-15. 2018.
> > [3] Li, Tian, Anit Kumar Sahu, Manzil Zaheer, Maziar Sanjabi, Ameet Talwalkar, and Virginia Smith. "Federated optimization in heterogeneous networks." International Conference on Machine Learning (ICML) workshop, pp. 1-6. 2019.”
> >
> > **Concrete use-case**
> >
> > “We appreciate this recommendation and we will make sure to include this when allowed to update. A typical use-case would be a cross-device setting where some critical model is being learned, such as, mapping to a battlefield scenario by integrating sensor data from multiple sensors or a smart farms scenario to integrate data from multiple sensor nodes.
> > In a sensitive setting, there could be a malicious intent to degrade the training process. Here an attacker could either compromise participating devices or join the training under multiple aliases. Having joined the network, it could intercept the communication between benign clients and the server, or directly compromise the server to get access to the benign gradients in order to craft the malicious gradients. This constitutes a full-knowledge attack. Further, prior work has shown that even the partial knowledge attack can be quite damaging (in such an attack, the adversary does not need to know the gradients of the benign clients, but assumes its own before-attack gradients to be an approximate estimate of the benign gradients).”

---

> > ### Author Response · Authors · 2021-11-23
> > **Mismatch in performance of FedSGSD, concrete use case, formatting**
> >
> > We have uploaded a revised version of the paper with changes suggested by the reviewer (Reviewer VCnh). Specifically these are the changes:
> >
> > 1. Discussion of the issue of mismatch in the performance of baseline FedSGD (Appendix A.4.3)
> >
> > 2. Discussion of concrete use-cases (Appendix A.4.1)
> >
> > 3. We replaced the word "Supplement" with "Appendix" as per the ICLR provided tex template. Section 8 (Reproducibility Statement) does not count toward the 9 page budget as per the author instructions.
> >
> > These changes we believe have improved the submission.

---

> > > ### Comment · Reviewer_VCnh · 2021-11-24
> > > **Relevant question**
> > >
> > > I appreciate the authors' efforts and I am willing to update my score.
> > >
> > > However, I would like to ask one final question to the authors: considering that Tesseract works also in the case of a "white box attacker" that knows how Tesseract operates, are the authors willing to claim that Tesseract is, in fact, a "Secure-by-Design" mechanism?
> > >
> > > If the authors are convinced of such statement (and the other reviewers do not find flaws with such statement), then inserting it in the paper would dramatically improve the magnitude of the contribution.

---

> > > > ### Author Response · Authors · 2021-11-25
> > > > **TESSERACT is secure-by-design in the following conditions**
> > > >
> > > > It is indeed secure by design to all untargeted model poisoning attacks that share the following characteristics:
> > > > 1. The adversaries are wishing to reduce the accuracy of the final global model and this is done by presenting incorrect gradient updates from the clients to the aggregation server (the most popular and logical way of launching model poisoning attacks) -
> > > >
> > > >        No matter what the attack mechanism is (directed-deviation attack is just one way in which the global model can be attacked), as long as an attack is untargeted, TESSERACT strikes at the root cause of poisoning by preventing high flip-score moves and preventing the model from going in the opposite direction to the direction toward the optima. We have made sure to take care of the iterations where the global model to be updated was itself poisoned in the previous iteration by trimming out low flip-score moves as well, as explained in Section 3, page 5, under “Flip-score”. Thus, by only allowing the clients to behave in a non-extreme behavior in terms of the relative flip-score, TESSERACT provides secure aggregation to a federated learning system.
> > > >
> > > > 2. The server is not compromised, only the clients are compromised -
> > > >
> > > >        Since TESSERACT is a part of the parameter server, the integrity of the server is a necessary requirement for the security of the entire federated learning system.
> > > > 3. The clients are not using homomorphic encryption in conveying their updates to the server -
> > > >
> > > >        Homomorphic encryption only allows mean aggregation at the parameter server, which is not byzantine-resilient. TESSERACT needs access to the clients’ gradients in order to compute the flip-score and decide whether or not a client is acting maliciously.

---

> > > > > ### Comment · Reviewer_VCnh · 2021-11-25
> > > > > **Good**
> > > > >
> > > > > Thanks for the response. Unless the other reviewers can confute such claims, then I will further raise my score.
> > > > >
> > > > > I strongly invite the authors to emphasize this characteristic **very early on** in the paper (abstract), and further remark it (i.e., Section I, Section III, and whenever it is necessary). From a security standpoint, a "secure-by-design" defense is very much appreciated in real world tasks.

---

### Decision · Program_Chairs · 2022-01-20

**Decision:**

Reject

**Comment:**

The paper presents a defense against the gradient sign flip attacks on federated learning. The proposed method is novel, technically sound and well evaluated. The crucial issue of the paper is, however, that this defense is specific to gradient-flip attacks. The authors show the robustness of their method against white-box attacks adhering to this threat model and claim that "an adaptive white-box attacker with access to all internals of TESSERACT, including dynamically determined threshold parameters, cannot bypass its defense". The latter statement does not seem to be well justified, and following the extensive discussion of the paper, the reviewers were still not convinced that the proposed method is secure by its design. The AC therefore feel that the specific arguments of the paper should be revised - or the claim of robustness further substantiated - in order for the paper to be accepted.

Furthermore, as a comment related to ethical consideration, the AC remarks that the paper's acronym, Tesseract, is used by an open source OCR software (https://tesseract-ocr.github.io/) as well as in a recent paper: Pendlebury et al., TESSERACT: Eliminating Experimental Bias in Malware Classification across Space and Time, USENIX Security 2019.

All of the above mentioned reservations essentially add up to a "major revision" recommendation which, given the decision logic of ACLR, translates into the rejection option.